# Conserved roles of *C. elegans* and human MANFs in sulfatide binding and cytoprotection

Meirong Bai[1], Roman Vozdek[1], Aleš Hnízda[2], Chenxiao Jiang[3], Bingying Wang[1], Ladislav Kuchar[4], Tiejun Li[5], Yuefan Zhang[5], Chase Wood[6], Liang Feng[6], Yongjun Dang[3] & Dengke K. Ma [ID] [1]

Mesencephalic astrocyte-derived neurotrophic factor (MANF) is an endoplasmic reticulum (ER) protein that can be secreted and protects dopamine neurons and cardiomyocytes from ER stress and apoptosis. The mechanism of action of extracellular MANF has long been elusive. From a genetic screen for mutants with abnormal ER stress response, we identified the gene *Y54G2A.23* as the evolutionarily conserved *C. elegans* MANF orthologue. We find that MANF binds to the lipid sulfatide, also known as 3-*O*-sulfogalactosylceramide present in serum and outer-cell membrane leaflets, directly in isolated forms and in reconstituted lipid micelles. Sulfatide binding promotes cellular MANF uptake and cytoprotection from hypoxia-induced cell death. Heightened ER stress responses of MANF-null *C. elegans* mutants and mammalian cells are alleviated by human MANF in a sulfatide-dependent manner. Our results demonstrate conserved roles of MANF in sulfatide binding and ER stress response, supporting sulfatide as a long-sought lipid mediator of MANF's cytoprotection.

[1] Cardiovascular Research Institute and Department of Physiology, University of California San Francisco, San Francisco, CA 94158, USA. [2] Institute of Organic Chemistry and Biochemistry, Academy of Sciences of the Czech Republic, 166 10 Prague 6, Czech Republic. [3] Key Laboratory of Molecular Medicine, Ministry of Education and Department of Biochemistry and Molecular Biology, School of Basic Medical Sciences, Shanghai Medical College, Fudan University, Shanghai 200032, China. [4] Research Unit for Rare Diseases, Department of Pediatrics and Adolescent Medicine, First Faculty of Medicine, Charles University and General University Hospital in Prague, Prague 12808, Czech Republic. [5] Department of Pharmacology, Second Military Medical University, Shanghai 200433, China. [6] Department of Molecular and Cellular Physiology, Stanford University, Stanford, CA 94305, USA. Meirong Bai and Roman Vozdek contributed equally to this work. Correspondence and requests for materials should be addressed to D.K.M. (email: Dengke.Ma@ucsf.edu)

O xidizing environment is essential for protein folding and functions in the endoplasmic reticulum (ER) of eukaryotic cells[1–3]. The Ero1-PDI enzymatic cascade in the ER mediates the oxygen-dependent disulfide bond formation and proper folding of transmembrane or secreted proteins. Severe hypoxia (reduced ambient oxygen), as frequently occurs in ischemic and neoplastic disorders, causes cellular ER stress response independently of the well-characterized hypoxia-inducible factor (HIF) pathway[4–6]. Eukaryotic cells actively maintain protein folding and redox homeostasis in the ER upon hypoxic and oxidative stresses by regulating numerous cellular factors to promote normal physiological functions and cytoprotection. Among such factors that have been studied, mesencephalic astrocyte-derived neurotrophic factor (MANF) is an enigmatic family of proteins with rather unique mode and mechanisms of action.

Mammalian MANF was initially identified for its neurotrophic effects on dopaminergic neurons[7]. It is a 20 kDa small secreted protein that exhibits no amino acid sequence similarities to any other classical families of target-derived neurotrophic factors, including glial cell-derived neurotrophic factor (GDNF) and brain-derived neurotrophic factor (BDNF) . MANF family proteins are highly evolutionarily conserved with both human and *Drosophila* orthologues being capable of promoting dopamine neuron survival[8–10]. Its N-terminus contains a structural fold similar to Saposin proteins while its C-terminus contains a SAP (SAF-A/B, Acinus, and PIAS) domain similar to that of Ku70, an inhibitor of proapoptotic Bax[11,12]. *MANF* is widely expressed in the body, with levels particularly high in the nervous system, kidney, heart, and pancreatic β cells[13–15]. Cellular MANF abundance and secretion are strongly up-regulated by ER stress-related stimuli, including hypoxia, ischemia, chemical treatments with tunicamycin or thapsigargin[13,15–17]. Because of its secretion from cardiomyocyte and autocrine/paracrine effects on cardiovascular functions, MANF has been termed "cardiomyokine" playing critical roles in cardioprotection, hypertrophy, and heart failure[18]. MANF KO mice are diabetic with abnormal activation of unfolded protein response in pancreatic islets, while its gain-of-function by exogenous MANF addition, physiological up-regulation, or transgenic MANF overexpression has been shown to protect dopaminergic neurons, cerebral neurons, Purkinje cells, retinal neurons, pancreatic β cells, and cardiomyocytes from degeneration or cell death[7,8,13,19–22].

Despite potent cytoprotective effects and emerging therapeutic potentials of MANF for neurodegenerative, cardiovascular, diabetic, and ischemic disorders, how extracellular MANF signals to cells and confers cytoprotection has remained largely elusive. No MANF receptor has been identified despite over a decade of research since its cytoprotective activity was discovered in 2003. Although it has been suggested that MANF might bind to certain cell surface lipids via its N-terminal Saposin-like domain, the molecular identity of MANF-binding lipids and how its potential lipid-binding ability might contribute to cytoprotection are unknown. Understanding the mechanisms of action of extracellular MANF in cytoprotection thus remains a key challenge of the field.

In this work, we started with a *C. elegans* genetic screen and identified an uncharacterized *C. elegans* orthologue of MANF as a regulator of the ER stress response. In a lipid biochemical screen, we discovered sulfatide as a direct interactor of both *C. elegans* and human MANFs. Sulfatide is a sulfoglycolipid synthesized in ER and Golgi, distributed to the extracellular leaflet of many cell types, and can be released to extracellular space and circulation in mammals[23,24]. By characterizing *C. elegans* and human MANFs in lipid binding and roles in protecting from ER stress and cell death, we provide multiple lines of evidence that MANF confers cytoprotection, at least in part, through direct binding to sulfatide and subsequent uptake by both *C. elegans* and mammalian cells.

## Results

**A genetic screen identifies a *C. elegans* manf-1 mutant**. We sought to identify new regulators of HIF-independent ER stress response by performing forward genetic screens for *C. elegans* mutants with abnormal ER stress responses. The *C. elegans* gene *hsp-4* encodes the orthologue of human ER chaperone BiP and is a well-established ER stress-inducible gene[25–28]. By random mutagenesis of the *C. elegans* transgenic strain in which GFP reporter expression is driven by the *hsp-4* promoter, we isolated mutants with constitutive GFP expression even without externally induced ER or hypoxic stresses. One mutation *dma1* caused fully recessive, completely penetrant, and strong constitutive activation of *hsp-4p::GFP*, predominantly in the intestine and pharyngeal epithelium of *C. elegans* (Fig. 1a). We genetically mapped the mutation *dma1* by single-nucleotide polymorphism-based linkage analysis and molecularly identified the causal gene (Supplementary Fig. 1). Based on RNA interference (RNAi) phenocopy of the *dma1* mutant and its transformation rescue with transgenic arrays of wide-type alleles, we determined that *dma1* defines a previously uncharacterized gene *Y54G2A.23*, which encodes the sole orthologue of the highly evolutionarily conserved MANF protein family (Fig. 1b, c, Supplementary Fig. 1). CRISPR-mediated deletions of *Y54G2A.23* caused constitutively high levels of *hsp-4p::GFP* with phenotype identical to that of *dma1* (Fig. 1c). We found that *hsp-4p::GFP* expression caused by *dma1* was suppressed by RNAi against *xbp-1* or *ire-1*, which encodes an ER stress sensor and activator of the ER stress-responding transcription factor XBP-1 (Fig. 1c). The mutation *dma1* as confirmed by Sanger sequencing is a missense transition mutation that converts serine 75 to leucine at the N-terminus of MANF-1 (named for Y54G2A.23; Fig. 1b, Supplementary Fig. 1d and 2).

*C. elegans manf-1* is widely expressed in several major tissues, including intestine, hypoderm, spermatheca, and nervous systems, based on an integrated transcriptional reporter in which *GFP* is driven by the endogenous *manf-1* promoter (Fig. 1d). The constitutive expression of *hsp-4p::GFP* in intestinal cells of *dma1* mutants was fully rescued by expression of *manf-1*(+) driven by the intestine-specific *ges-1* promoter, the ubiquitously active *rpl-28* promoter or a transgenic array expressing human *MANF* driven by the *C. elegans manf-1* promoter (Fig. 1e, f). By contrast, a mutant *manf-1*(C146G) carrying the substitution of an invariant cysteine C146 completely conserved in the MANF protein family failed to rescue the constitutive *hsp-4p::GFP* expression of *dma1* mutants (Fig. 1e). In addition, we found that chronic (24 h) but not acute (1 h) exposure of *manf-1* mutants to tunicamycin, an ER stress-inducing protein glycosylation inhibitor, decreased progeny numbers to a larger degree in *manf-1* mutants than wild type (Supplemental Fig. 1e). These findings indicate that *C. elegans manf-1* plays an evolutionarily conserved role in regulating ER stress responses.

**Both *C. elegans* and human MANFs bind directly to sulfatide**. To explore novel mechanisms of MANF's cytoprotection, we first examined structural motifs of the hitherto uncharacterized *C. elegans* MANF-1 (Ce-MANF). Based on homology-based structural modeling against the crystal structure of Hs-MANF[29], Ce-MANF contains a saposin-like fold at the N-terminus and a SAP-like C-terminal domain (Supplementary Fig. 2a, b). Since Saposin family proteins solubilize various sphingolipids for degradation in lysosomes[30], we tested the binding of purified Ce-MANF to an

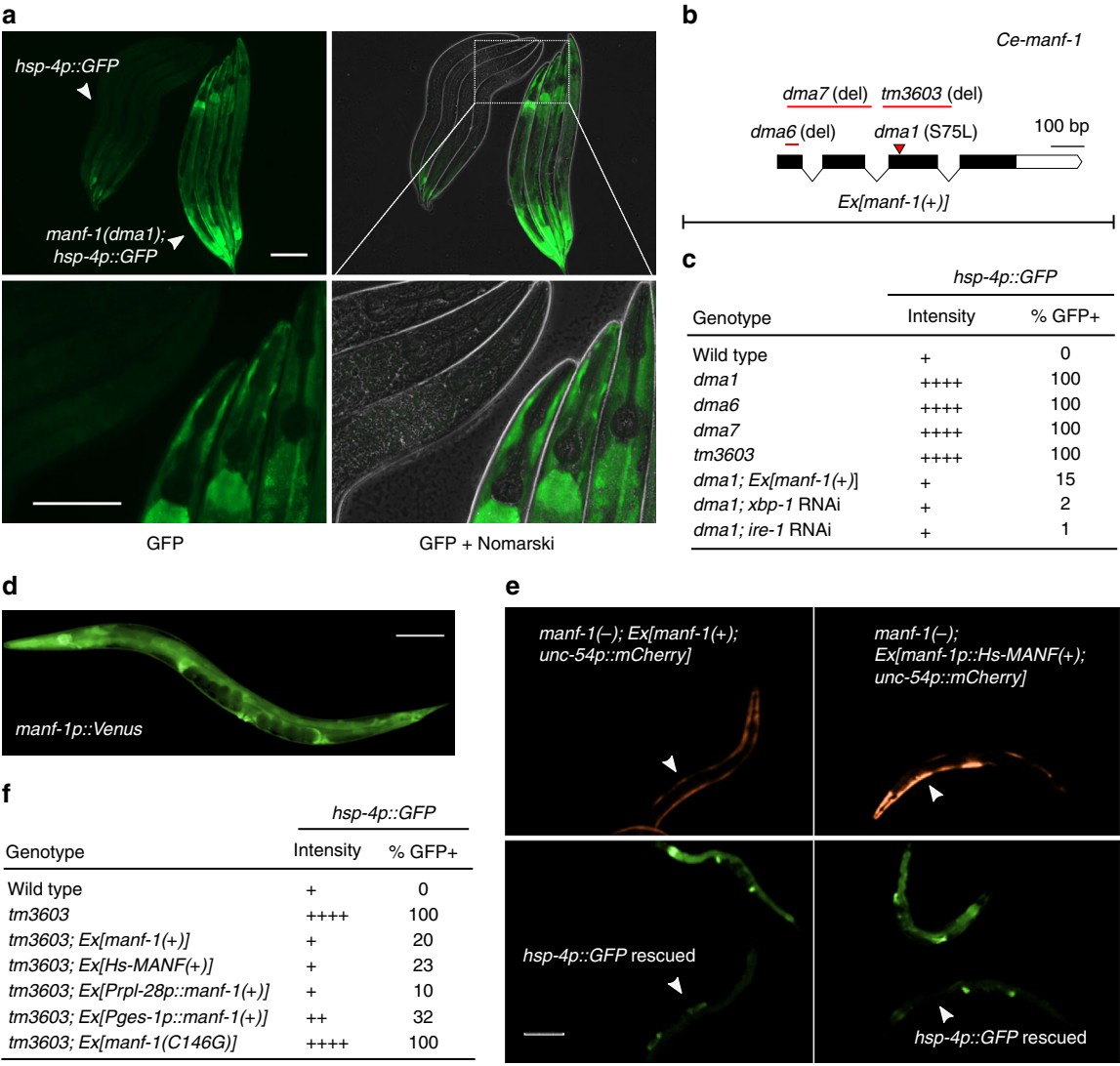

**Fig. 1** A genetic screen identifies *C. elegans manf-1* in regulating *hsp-4p::GFP*. **a** Exemplar GFP fluorescence and Nomarski images showing constitutive activation of *hsp-4p::GFP* in the *dma1* mutant isolated from mutagenesis screens. Wild type and mutant L4-stage animals (indicated by arrows) are aligned in the same field to compare GFP intensity at low (above) or high (bottom) magnification. Scale bars, 100 μM. **b** Schematic of *manf-1* gene structure with various *manf-1* alleles and extrachromosomal arrays (Ex) indicated. **c** Summary table for the *hsp-4p::GFP* phenotype of the wild type compared with various mutants, with indicated levels of GFP intensity in the intestine, the major site of *hsp-4p::GFP* expression, and penetrance ($N \geq 100$ for each genotype). **d** Integrated *manf-1p::Venus* transcriptional reporter indicating the widespread expression of *manf-1*. **e** Exemplar GFP fluorescence images showing the rescue of the *hsp-4p::GFP* phenotype in the *manf-1* null *tm3603* mutant by *C. elegans manf-1(+)* and human *MANF* driven by the *manf-1* promoter. Transgenic rescuing arrays are marked by *unc-54p::mCherry* expressed in body wall muscles. **f** Summary table showing the rescuing effect of various transgenes on *hsp-4p::GFP* of the *manf-1(tm3603)* mutant with indicated GFP intensity levels and penetrance ($N \geq 100$ for each genotype)

array of 16 structurally defined sphingolipids spotted on a nitrocellulose membrane (Echelon sphingostrip lipid overlay assay). We found that Ce-MANF binds specifically to one single lipid species, corresponding to sulfatide (Fig. 2a). Purified Hs-MANF also exhibited binding to sulfatide, prepared from either bovine brains or synthetic sources (Fig. 2b, Supplementary Fig. 2e). Using lipid overlay assays, we examined other lipids that are structurally similar to sulfatide but otherwise lacking or with variant chemical groups and found that both the acyl chain and sulfate are essential for the binding (Fig. 2c). To determine whether MANF can bind to sulfatide freely in solution we performed size exclusion chromatography of micelle-reconstituted (by Tween-20 and dodecyl maltoside detergents) Hs-MANF with or without sulfatide. We observed association of Hs-MANF with detergent micelles only when sulfatide was present (Fig. 2d, e). In

addition, sulfatide specifically deterred enzymatic digestion of Hs-MANF by trypsin (Fig. 2f, Supplementary Fig. 3a, b) and protected Hs-MANF against thermal denaturation (Supplementary Fig. 3c). These results identify sulfatide as a sulfated glycolipid that directly binds to Ce-MANF and Hs-MANF.

**Sulfatide binding mediates conserved cytoprotection of MANF.** Since MANF protein family members are highly evolutionarily conserved (Supplementary Fig. 2c, f), we investigated whether MANF binding by sulfatide might facilitate cytoprotection of MANF in both *C. elegans* and mammalian cells.

In *C. elegans*, we found that the constitutively high expression level of *hsp-4::GFP* in *manf-1* mutants can be fully rescued by transgenic arrays expressing either wild-type *C. elegans manf-1* or

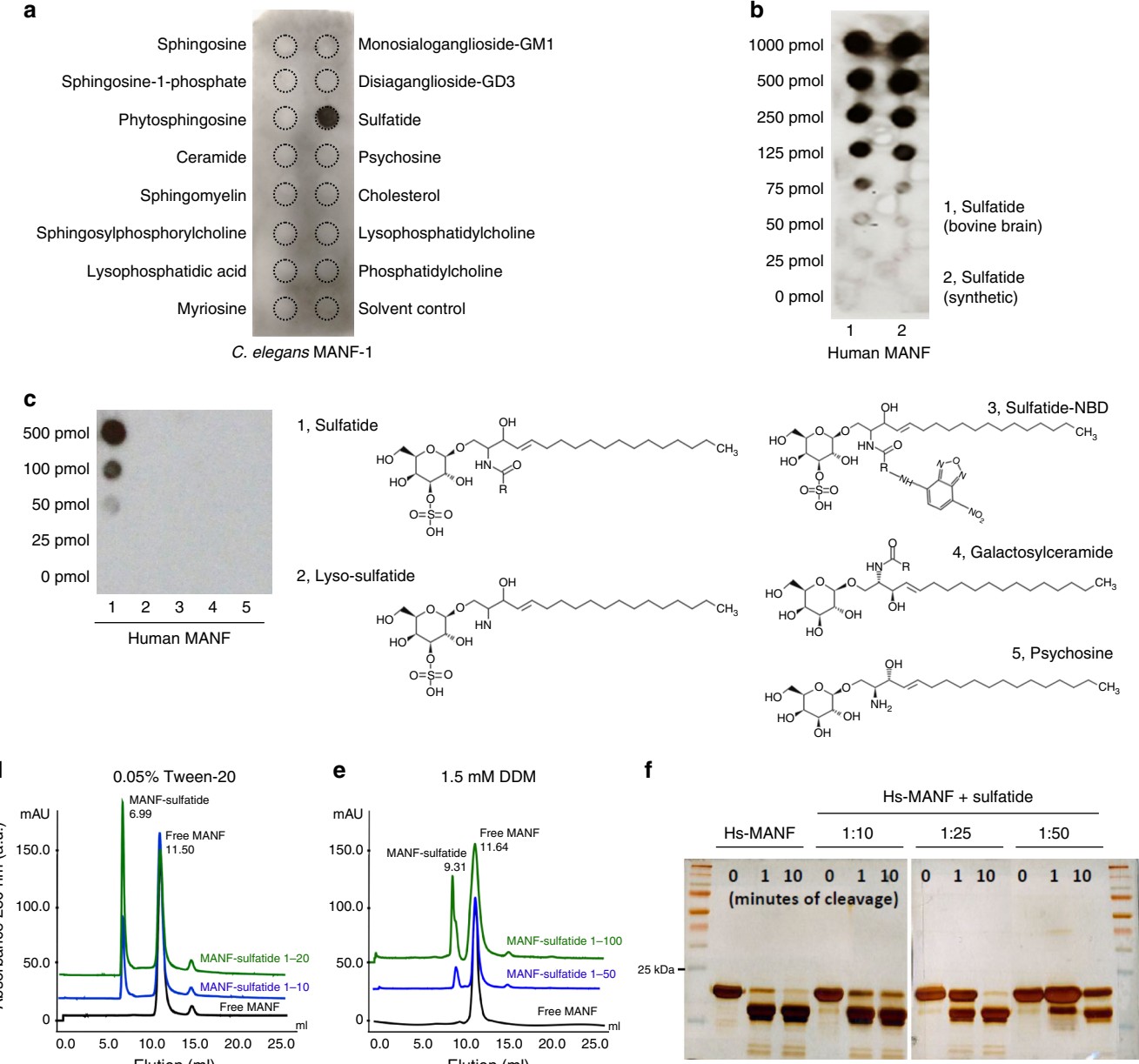

**Fig. 2** Purified MANF binds to sulfatide. **a** Lipid overlay assay with an Echelon Sphingostrip membrane showing specific binding of His-tagged Ce-MANF to sulfatide. **b** Lipid overlay assay with customized lipid membrane showing dose-dependent binding of His-tagged Hs-MANF to sulfatide, prepared from bovine brains or synthetic source. **c** Lipid overlay assay with customized lipid membrane spotted with sulfatide-related lipids showing that sulfate as well as both acyl chains moieties are required for the sulfatide::Hs-MANF interaction. **d** Size exclusion chromatography with Hs-MANF reconstituted in 0.05% Tween-20 and varying doses of sulfatide. Molar ratio of MANF:Sulfatide at 1:20 was sufficient for 50% fraction to be bound onto tween-sulfatide micelles (peak at 6.99 ml elution). **e** Size exclusion chromatography with Hs-MANF reconstituted in 1.5 mM dodecyl maltoside (DDM) and varying doses of sulfatide. Molar ratio of MANF:Sulfatide at 1:100 resulted in 40% protein fraction to be bound onto DDM-sulfatide micelles (peak at 9.31 ml elution). **f** Limited proteolysis using trypsin protease showing that Hs-MANF is protected against digestion upon sulfatide binding, under conditions of different cleavage times (0, 1, and 10 min) and MANF versus sulfatide molar ratios (1:10, 1:25, and 1:50)

human *MANF* (Fig. 1c, e, f). We generated HEK293T lentiviral stable cell lines that constitutively express and secrete Hs-MANF, which carries a C-terminal V5 epitope for tagging and to minimize ER retention[31]. Using the secreted Hs-MANF collected from cell culture media, we found that *manf-1* null *C. elegans* mutants can be partly rescued by incubation with exogenous Hs-MANF protein in the presence of sulfatide in liquid culture (Fig. 3a–c). The reduction of *hsp-4*::GFP levels was accompanied with increased levels of Hs-MANF uptake in *C. elegans manf-1* null mutants (Fig. 3c). The

uptake of Hs-MANF in *C. elegans* was mediated, at least in part, by specific dynamin-dependent endocytosis machinery (Fig. 3d) in *C. elegans* and was independent of epitope tagging, bacterial food presence or stages of animals but dependent on presence of sulfatide (Supplementary Figs. 3d and 5c). The uptaken Hs-MANF appeared widely localized in *C. elegans* tissues (Fig. 3e). These results indicate that Hs-MANF can by uptaken in a sulfatide-dependent manner and rescue the ER stress response of *manf-1* null mutants in *C. elegans*.

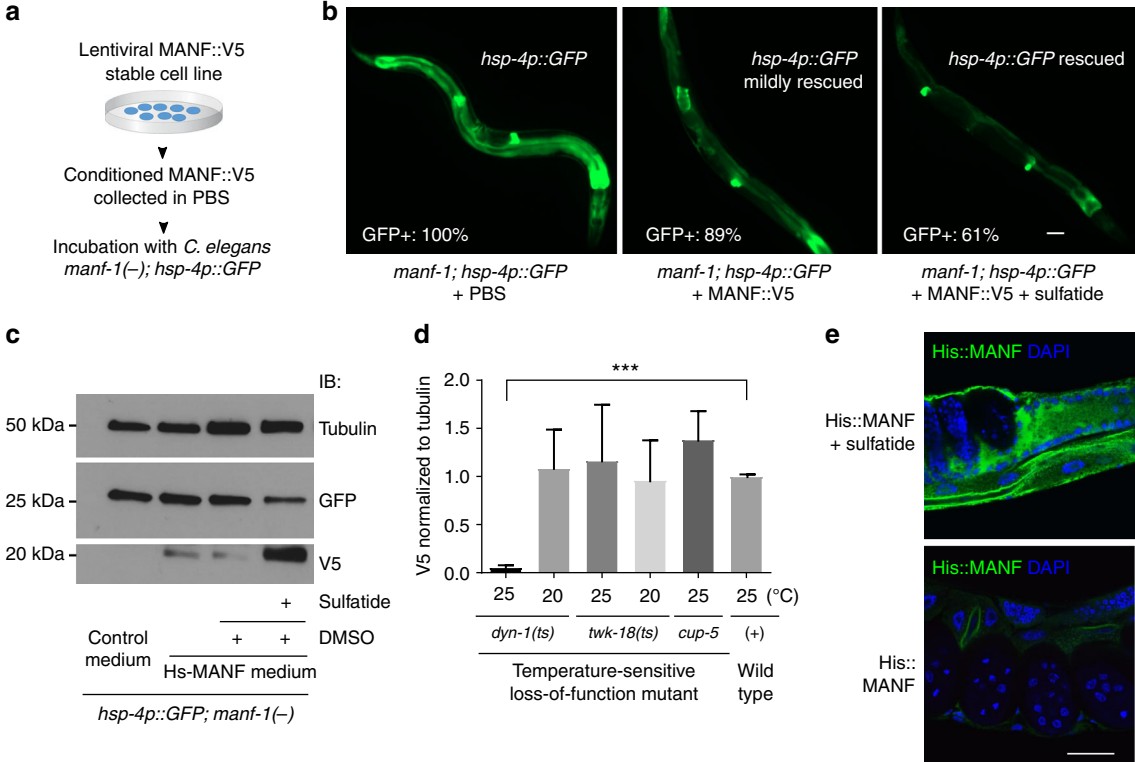

**Fig. 3** ER stress-rescuing effects of MANF and sulfatide in *C. elegans*. **a** Schematic of experimental workflow to test rescuing effects of human MANF from conditioned cell culture PBS. **b** Exemplar GFP fluorescence images showing the rescue of the *hsp-4p::GFP* phenotype in the *manf-1* null *tm3603* mutant by incubation with purified MANF secreted from conditioned media. MANF alone rescued only mildly compared with fuller rescue by MANF plus exogenous sulfatide addition (500 μM in DMSO; sulfatide alone had no rescue), with indicated penetrance of rescue (N ≥ 100 for each genotype). **c** Exemplar western blot of Hs-MANF in *C. elegans manf-1* mutants after incubating with Hs-MANF, showing sulfatide dependency of endocytosis and rescue of *hsp-4*::GFP levels by endocytosed Hs-MANF. **d** Normalized levels of Hs-MANF::V5 immunoblot signals (from three biological replicates of N ≥ 100 animals) showing block of Hs-MANF endocytosis in temperature-sensitive endocytosis mutant *dyn-1*(ts) cultured from 25 °C but not at the permissive temperature, 20 °C. **e** Exemplar immunofluorescence images of *C. elegans* tissues stained with anti-His (green) and DAPI (blue) indicating the uptaken His::MANF with sulfatide. Scale bar, 15 μm

We next examined effects of MANF and sulfatide in cultured mammalian cells. MANF::V5 from the conditioned medium of the lentiviral stable line can bind to sulfatide (Fig. 4a) and protect both HEK293T cells as well as the H9C2 cardiomyocyte cell line after severe ER stress or hypoxia with glucose deprivation conditions (Supplementary Fig. 4a–c). To probe the specific role of sulfatide binding in conferring cytoprotection, we compared the effect of wild-type Hs-MANF with a K112L mutant exhibiting reduced binding activity to sulfatide. We found that sulfatide did not bind to the Hs-MANF paralogue cerebral dopamine neurotrophic factor (CDNF) (Supplementary Fig. 2e). The evolutionarily conserved surface lysine residue K112 in Hs-MANF (K88 in crystal structure of Hs-MANF, substituted with leucine in CDNF) partly accounts for the difference between MANF and CDNF[29]. We found that the MANF (K112L) mutant exhibited markedly reduced binding to sulfatide, without affecting expression levels (Fig. 4a–c). The K112L mutation attenuated the survival-promoting effect of Hs-MANF in both HEK293T and H9C2 cells (Fig. 4d, e, Supplementary Fig. 4c). Cytoprotective effects of Hs-MANF was also blocked by the O4 monoclonal antibodies against sulfatide, further demonstrating the essential role of endogenous sulfatide for cytoprotection of Hs-MANF (Fig. 4d, e). These results indicate that potent cytoprotective effects of Hs-MANF are facilitated by specific binding to sulfatide.

**Binding to sulfatide promotes cellular uptake of MANF.** Given the abundance of cell surface and extracellular sulfatide as well as its known roles in membrane trafficking[23,24], we hypothesize that sulfatide facilitates MANF functions by promoting cellular uptake of MANF both in *C. elegans* (Fig. 3c–e) and mammalian cells. Indeed, we found that secreted wild type but not K112L mutant Hs-MANF proteins from conditioned media were efficiently uptaken by target HEK293T cells in a dose-dependent manner (Fig. 5a, b). The uptaken Hs-MANF appeared to extensively co-localize with the MANF-interacting protein GRP78 (i.e. the ER resident chaperone BIP), indicating its likely endocytosis following cell surface binding (Fig. 5c). The uptake of wild-type Hs-MANF can be further enhanced by exogenous sulfatide or increased endogenous levels of sulfatide in target cells by over-expressing the gene encoding the sulfatide-biosynthetic enzyme cerebroside sulfotransferase (CST) (Fig. 5a, Supplementary Fig. 5a). To test whether MANF might be endocytosed to promote the intracellular degradation of sulfatide, as the canonic saposin-B protein does, we measured sulfatide levels in Hs-MANF over-expressing and CRISPR-mediated KO cells and found Hs-MANF does not affect sulfatide levels (Supplementary Fig. 5b). As the V5-tagged MANF, the tag-free MANF was similarly uptaken by target cells in a sulfatide-dependent manner (Supplementary Fig. 5c). These results indicate that MANF binds to sulfatide for efficient cellular endocytosis, not for degradation.

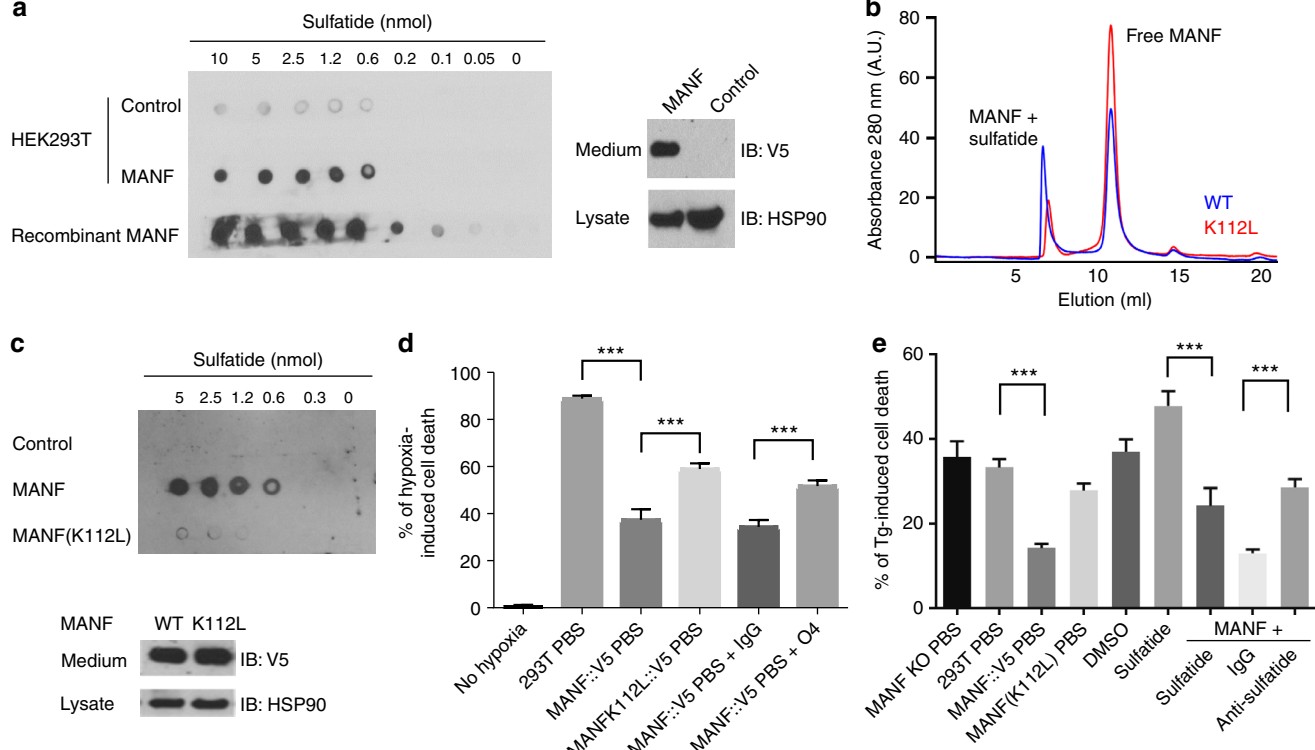

**Fig. 4** Sulfatide binding is critical for cytoprotection by MANF in culture. **a** Lipid overlay assay showing secreted V5-tagged Hs-MANF and recombinant His-tagged Hs-MANF bound to sulfatide (left), with secreted Hs-MANF::V5 confirmed by western blot of cell lysate and medium (right). HSP90 was used as a loading control. **b** Size exclusion chromatography of wild-type Hs-MANF and K112L mutants reconstituted in 0.05% Tween-20 and sulfatide. Only 20% of the K112L mutant Hs-MANF was bound onto the micelle fraction compared with 50% of the wild-type Hs-MANF. **c** Lipid overlay assay (above) showing that secreted MANF (K112L) from HEK293T cell lines exhibited markedly reduced binding to sulfatide with western blot (bottom) showing normal protein levels in the HEK293T stable line. **d** Death rates of HEK293T cells after 48 h of 0.1% hypoxia with pretreatment of Hs-MANF or MANF (K112L) mutant-conditioned media or with additional anti-sulfatide O4 antibodies. **e** Death rates of HEK293T cells after 24 h of indicated treatment followed by 40 h exposure to 0.5 μM thapsigargin (Tg). Error bars: S.E. with ***$P < 0.001$ ($N \geq 3$ independent biological replicates)

We next characterized the role and mechanism of action of the sulfatide-mediated enhancement of Hs-MANF endocytosis. Endogenous sulfatide is biosynthesized in ER and Golgi while MANF is normally retained in the ER and up-regulated by ER stress and hypoxia, followed by secretion and cellular release through the ER/Golgi pathway[16,19]. We verified that severe hypoxia can indeed drastically increase the abundance of Hs-MANF into the extracellular medium while its secretion can be blocked by brefeldin A, a classic inhibitor of the ER/Golgi secretion pathway (Supplementary Fig. 6a, b). In addition, we found that the ratio of bound/free MANF increases at pH 6.0 and 6.75 (Supplementary Fig. 6c), characteristic of the Golgi lumen. These results suggest that sulfatide is likely associated with MANF during its secretion and facilitates the subsequent endocytosis of MANF to target cells.

To minimize the contribution from endogenous and other sources of sulfatide present in the cell culture medium that contains serum and thus sulfatide, we collected the secreted Hs-MANF in phosphate-buffered saline (PBS) and found that adding exogenous sulfatide to Hs-MANF in PBS markedly increased Hs-MANF endocytosis in a dose- and time-dependent manner (Supplementary Fig. 7a–c). Furthermore, treatment with sulfatide antibodies or its degrading enzyme sulfatase in the cell culture medium attenuated Hs-MANF endocytosis (Supplementary Fig. 7a). To exclude the potential involvement of other factors in the conditioned medium indirectly affecting endocytosis, we purified Hs-MANF from cell culture media using affinity antibodies against the V5

epitope in Hs-MANF and found that antibodies against sulfatide inhibited endocytosis while exogenous sulfatide drastically enhanced endocytosis (Supplementary Fig. 7b). The enhancement of Hs-MANF endocytosis by sulfatide was apparently specific since sulfatide exhibited no enhancing effects on the endocytosis of another two distinct secreted proteins (Supplementary Fig. 7d), and can be blocked by inhibitors of endocytic machineries[32] (Supplementary Fig. 7e). These results provide further evidence that the sulfatide level is a critical and specific determinant of Hs-MANF endocytosis.

**MANF-sulfatide alleviates ER stress response in cultured cells.** C. elegans experiments clearly indicated the roles of both Ce-MANF and Hs-MANF in alleviating ER stresses (Figs. 1 and 3). To further examine whether MANF can alleviate ER stresses also in mammalian cells, we monitored the formation of G3BP-marked stress granules in U2OS cells[33,34] evoked by thapsigargin (Tg), an ER stress triggering inhibitor of mammalian sarco/endoplasmic reticulum Ca²⁺-ATPase. We found that Tg induced a time-dependent stress granule formation that were markedly attenuated by pretreatment with Hs-MANF (Fig. 6a). Compared with Hs-MANF, the Hs-MANF K112L mutant exhibited markedly reduced ability to attenuate stress granule formation (Fig. 6b, c). The effect of exogenous sulfatide manifested more prominently with lower doses of Hs-MANF, suggesting that endogenous sulfatide bound to Hs-MANF affected stress granule

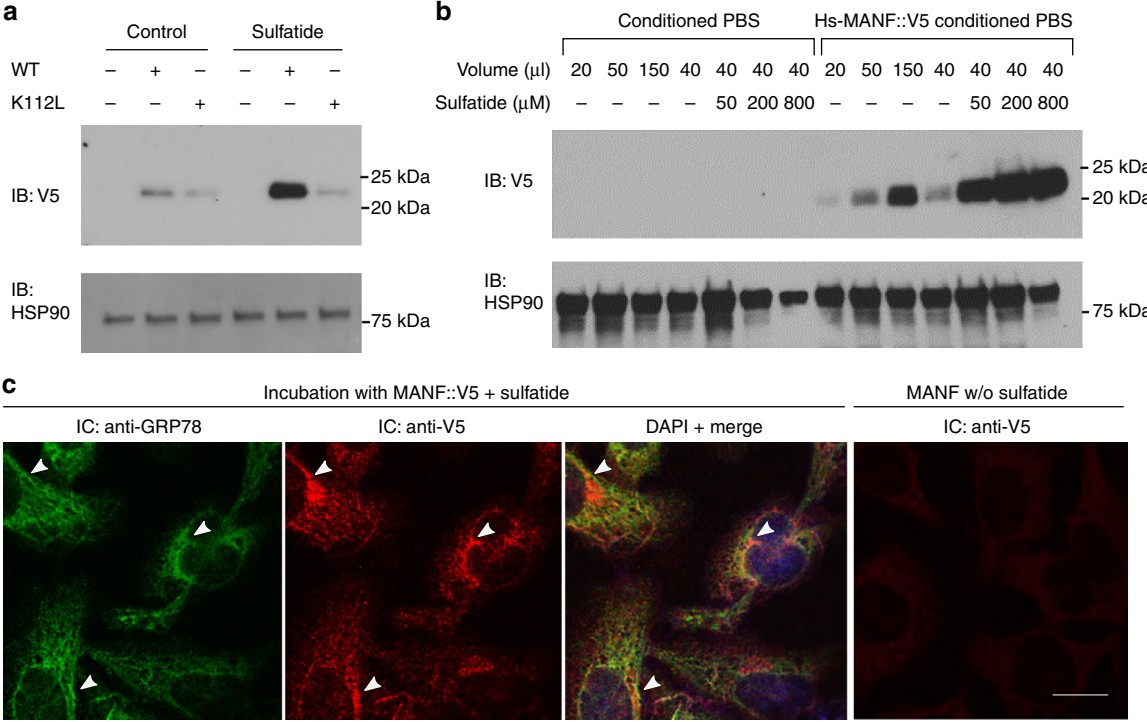

**Fig. 5** Sulfatide promotes cellular uptake of MANF. **a** Exemplar western blot of lysates from target HEK293T cells treated with Hs-MANF or Hs-MANF (K112L) (purified from conditioned PBS of HEK293T-MANF::V5 cell lines) with or without sulfatide. **b** Exemplar western blot of lysates from target HEK293T cells treated with Hs-MANF (purified from conditioned PBS of HEK293T-MANF::V5 cell lines) with increasing doses of sulfatide added to the target HEK293T cell medium. **c** Exemplar immunofluorescence images of endogenous GPR78 (green) and V5-stained Hs-MANF (red), merged image with DAPI after incubation with Hs-MANF (purified from conditioned PBS of HEK293T-MANF::V5 cell lines) and with V5 staining of no-sulfatide control. Arrow heads indicate representative loci of co-localization. Scale bar, 20 μm

formation (Fig. 6c). These results support the notion that sulfatide-bound MANF alleviates ER stress responses also in mammalian cells.

## Discussion

The cytoprotective and ER stress-alleviating effects of MANF are highly evolutionarily conserved[8–10,13]. In mammals, MANF has been shown to be particularly effective in protecting dopamine neurons, retinal ganglion cells, and cardiomyocytes against oxidative and ER stresses[8,16,19]. However, molecular mechanisms of how secreted MANF elicits cytoprotection have been undefined. Our studies identify sulfatide as a lipid interactor of MANF and the sulfatide-binding capacity is critical for cytoprotective effects of MANF. We further demonstrate that sulfatide promotes cellular uptake of MANF in both *C. elegans* and mammalian cells and that *C. elegans manf-1* mutants can be rescued by human MANF, suggesting highly evolutionarily conserved mechanisms by which MANF alleviates ER stress and cell toxicity under hypoxic and ER stress conditions.

Both Ce-MANF and Hs-MANF contain saposin-like structural folds at N-termini (Supplementary Fig. 2)[13]. Limited proteolysis by trypsin digestion followed by mass spectrometry identified predominantly the N-terminal part of Hs-MANF protected upon sulfatide addition, suggesting that the N-terminal saposin-like domain of MANF binds to sulfatide directly (Supplementary Fig. 3a, b). Hypoxia, which lowers pH of the extracellular environment, promotes MANF secretion (Supplementary Fig. 6b) and likely enhances cytoprotection against hypoxic toxicity. Several other saposin-like proteins have been shown to bind to sphingolipids; in particular, saposin-B binds to and presents sulfatide

to arylsulfatase A for sulfatide degradation in the lysosome[30]. Unlike saposin-B, however, MANF does not affect sulfatide levels (Supplementary Fig. 5b). Our findings suggest that MANF retains the sulfatide-binding capacity by the conserved saposin-like structural fold and is molecularly chaperoned by sulfatide, rather than presenting sulfatide for degradation, through the secretion pathway and extracellular milieu to facilitate eventual cellular uptake and cytoprotection in target cells.

Sulfatide-dependent cellular uptake of MANF also suggests a previously unappreciated mechanism of its neurotrophic activity. Based on collective evidence from *C. elegans* genetics, in vitro lipid–protein biochemistry and mammalian cell culture model, we propose that extracellular sulfatide-bound MANF enters the cell through endocytosis and mediates cytoprotection by promoting ER homeostasis and/or interacting with additional cytoprotective effectors. Our endocytosis model can thus reconcile the established effect of extracellular MANF and its reported intracellular effect in cytoprotection, e.g. by direct binding and inhibition of proapoptotic Bax family proteins[11,12]. Whether sulfatide binding of MANF requires additional cell surface receptor(s) and mechanisms how their interaction elicits intracellular signaling events to mediate cytoprotection await further investigation.

Sulfatide is biosynthesized first in the ER where the ceramide moiety is converted to galactocerebroside and subsequently sulfated to generate sulfatide by CST in the Golgi. Arylsulfatase A (ARSA) is activated by saposin-B and degrades sulfatide in the lysosome. Intriguingly, the *C. elegans* genome appears to encode a homolog (*sul-2*) of ARSA but not CST. We speculate that *C. elegans* might take sulfatide from its feeding bacteria to facilitate MANF signaling, although it remains also possible that *C. elegans* MANF relies on endogenous sulfatide-like lipids for trafficking

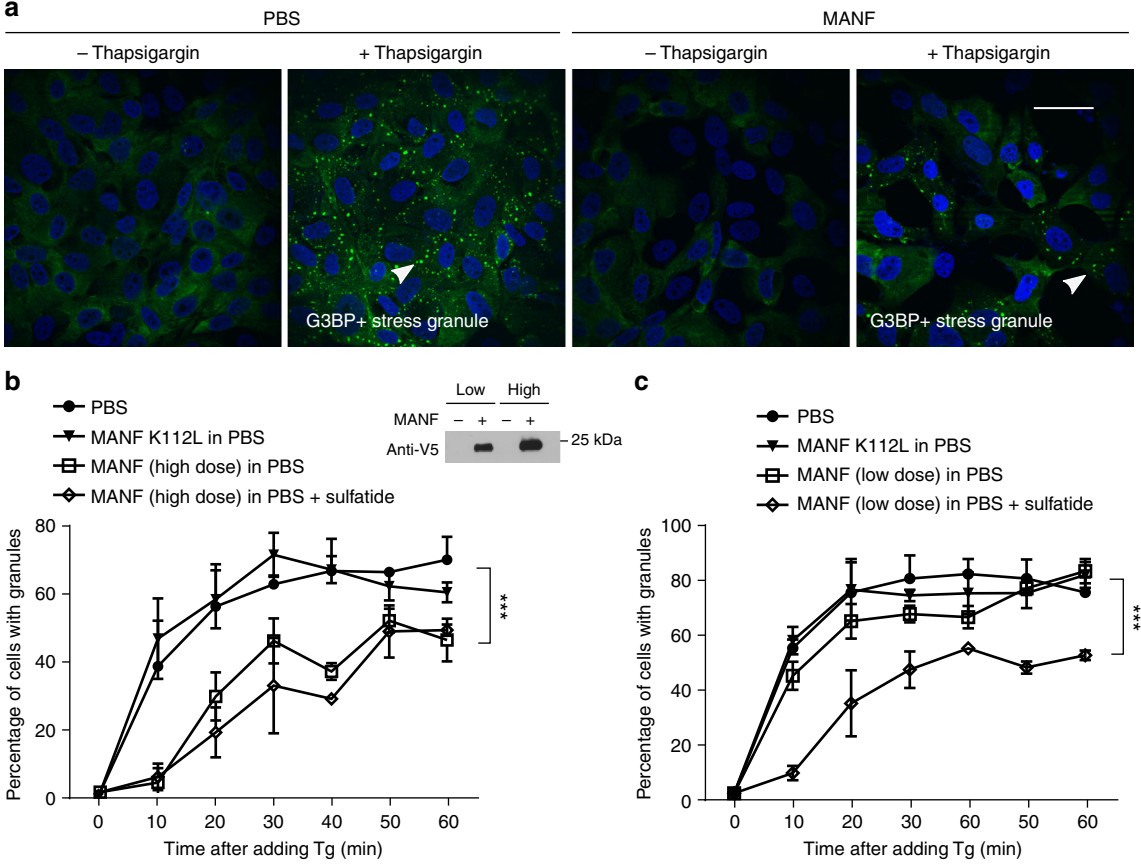

**Fig. 6** Sulfatide-bound MANF alleviates ER stress in mammalian cells. **a** Exemplar confocal GFP fluorescence images of Tg-induced stress granules in U2OS cells pretreated with conditioned MANF in PBS or control PBS. Scale bar, 40 μm. **b** Quantification of the percentages of cells with granules induced by Tg in 60 min and high doses (10 μl of conditioned medium/well) of Hs-MANF. A western blot for V5-tagged Hs-MANF indicates low and high doses of Hs-MANF used for experiments in **b** and **c**. **c** Quantification of the percentages of cells with granules induced by Tg in 60 min and low doses (2 μl of conditioned medium/well) of Hs-MANF. Cells were deprived of serum for 48 h and then treated with conditioned PBS or MANF for overnight. Plots represent quantification of the data from 10 images per condition from three independent experiments. Scale bar, 15 μm. *** indicates $P < 0.001$

and paracrine effects across tissues. In the mammalian nervous system, sulfatide is present primarily on the extracellular leaflet of the oligodendroglia-derived plasma membrane. However, sulfatide is also present in extracellular matrix, blood serum, and on the extracellular membrane leaflet of many other cell types. Thus, it is conceivable that sulfatide as a lipid chaperone facilitates MANF traveling across cells and also as a cell-surface lipid-type receptor mediates the paracrine cytoprotection of MANF in vivo. Aberrant levels and metabolism of sulfatide cause metachromatic leukodystrophy characterized by severe neuropathy and are associated with Parkinson's disease, Alzheimer's disease, cardiovascular disorders, and cancer[24,35]. The discovered physical interaction between MANF and sulfatide as well as the functional importance of their interaction for cytoprotection should help understand their respective roles in physiology and therapeutic potentials in many pathophysiological settings.

## Methods

***C. elegans* strains and assays**. *C. elegans* strains were cultured with standard procedures unless otherwise specified. The N2 Bristol strain was used as the reference wild-type strain, and the polymorphic Hawaiian strain CB4856 was used for genetic mapping and SNP analysis[36,37]. Forward genetic screen for constitutive GFP reporter-activating mutants after ethyl methanesulfonate (EMS)-induced random mutagenesis was performed as described previously[38,39]. To generate *manf-1* null alleles in *C. elegans*, we used CRISPR-Cas9 to induce double-stranded breaks and subsequent non-homologous end joining event caused various deletions of the *manf-1* gene and constitutive activation of *hsp-4p*::GFP. Feeding RNAi

starting from larval L1 stages was performed as previously described[40]. Transgenic strains were generated by germline transformation as described[41]. Transgenic constructs were co-injected (at 10–50 ng/μl) with *unc-54p*::mCherry, and lines of mCherry+ animals were established. Strains used are as follows: *zcIs4, manf-1 (dma1) IV; zcIs4 V, manf-1(tm3603) IV; zcIs4 V, manf-1(dma6) IV; zcIs4 V, manf-1 (dma7) IV; zcIs4 V, zcIs4 V, dyn-1(ky51) X; twk-18(cn110) X; dmaEx42[manf-1(+), unc-54p::mCherry]; dmaEx28[manf-1p::Hs-MANF(+), unc-54p::mCherry]; dmaEx115[rpl-28p::manf-1::Venus], dmaIs4[manf-1p::GFP]*.

To examine effects of tunicamycin on progeny numbers and development, well-fed L4 larvae were placed on the NGM plate containing 50 μg/ml of tunicamycin for 1 or 24 h and then moved to fresh NGM plates without tunicamycin. Numbers of viable progeny were counted in 24-h intervals. Five biological replicates (two founders per plate) were used for statistical analysis.

**Cell culture**. HEK293T, H9C2, U2OS, and HeLa cell lines were initially purchased from the American Type Culture Collection (ATCC, Manassas, VA, USA). The cells were routinely cultured in Dulbecco's modified Eagle's medium (DMEM) supplemented with 10% (v/v) fetal bovine serum (FBS) and 1% penicillin/strep-tomycin at 37 °C with 5% carbon dioxide. For in vitro cell model of H/R, oxygen-glucose deprivation in 293T or H9c2 cells was induced by replacing the complete high-glucose DMEM to glucose and serum-deprived medium in a hypoxia incubator chamber (Nuaire) saturated with 99% nitrogen (modified from ref.[42]). After 5 or 16 h of hypoxia, the cells were cultured under no glucose condition and subjected to reoxygenation under normoxia (21% $O_2$) for 8 h.

MANF knockout HEK293T cells were generated using the CRISPR-Cas9 system as previously described[43]. Briefly, the online design tool (http://crispr.mit.edu/) was used to select two high-score guide RNA (gRNA) target sites located at first and the last exons containing a 20-base pair (bp) target sequence (targeting sequences were GCGGTTCAGTCGGTCGGCGG and GTGCACGGACCGATT TGTAG). Two pairs of 24- or 25-mer oligos for each guide were designed with *Bbs*I restriction enzyme cutting site. The oligos were annealed and phosphorylated, then

cloned into pSpCas9 (BB) plasmid (pX330) (Addgene plasmid ID 42230). HEK293T cells were transiently transfected with two CRISPR/Cas9 construct targeting MANF and puromycin expression construct. The next day, cells were selected with puromycin for 2 days and subcloned to form single colonies. MANF KO clones were identified by PCR screen. The obtained clones were validated by qPCR and immunoblot using an antibody against MANF.

**DNA constructs and lentiviral transfection**. The pLX304-MANF and GAL3ST1 (also known as CST) plasmids were purchased from DNASU (Tempe, AZ, USA) and the pLX304-MANF (K112L) was generated using Q5 site-directed mutagenesis (New England Biolabs, Beverly, MA, USA). The tag-free pLX304-MANF was generated using Q5 site-directed mutagenesis to remove the sequence encoding V5 tag. *C. elegans* constructs for transgenic strains were generated by direct PCR and PCR fusions (plasmids and primers available upon request).

For lentiviral transfection, the plasmids with packaging plasmids were co-transfected into HEK293FT (with a ratio of 2:1.5:1.5) using Turbofect reagent (Thermo Fisher Scientific Inc. Waltham, MA, USA) according to the manufacturer's instructions. Lentivirus-containing medium was filtered from the post-transfection supernatant and used for transduction of HEK293T cells. All lentivirus-infected cells were cultured in the medium containing Polybrene (8 μg/ml) from Sigma Aldrich (St. Louis, MO, USA). Forty-eight hours after transduction, the cells were selected with 15 μg/ml Blasticidin S (Thermo Fisher Scientific Inc., Waltham, MA, USA).

**Purification MANF::v5 from conditioned medium**. Stably MANF::V5-expressing HEK293T cells were washed three times with PBS and incubated with PBS at 37 °C for 4 h. The conditioned PBS was centrifuged at 2500 rpm for 10 min and filtered. Then the conditioned PBS was concentrated using Amicon Ultra-15 10K centrifugal filter devices for 100-fold concentration according to the manufacturer's instructions. For purification of MANF::V5 from conditioned PBS, the Anti-V5 affinity gel (Sigma Aldrich, St. Louis, MO, USA) was thoroughly suspended and 80 ul of the gel suspension was transferred to a fresh tube with addition of 1 ml of PBS thoroughly suspended by pipetting followed by washing with PBS for four times. Concentrated conditioned PBS was added to the washed resin and gently agitated for 2 h at 4 °C. The resin was centrifuged for 30 s at 3000 rpm, washed with 1 ml ice-cold PBS for five times. After the final wash, 100 μl of elution buffer (0.1 M glycine HCl, pH 3.5) was added to each sample, which was incubated with gentle shaking for 5 min at room temperature. The resin was centrifuged for 5 s at 8200 $g$ and the supernatants were immediately transferred to fresh tubes containing 10 μl of 0.5 M Tris-HCl, pH 7.4, 1.5 M NaCl, to neutralize samples.

**Immunoblot analysis**. Laemmli loading buffer (Bio-Rad Lab, Hercules, CA, USA) plus 5% β-mercaptoethanol was added to cell pellets or worm pellets before heating at 95 °C for 5–15 min. Around 30 μg of whole cell protein lysate samples were separated on mini-PROTEIN GTX precast gels, and transferred to nitrocellulose membranes (Bio-rad Lab, Hercules, CA, USA). After blocking in TBS containing 5% non-fat milk and 0.1% Tween-20, the membranes were incubated overnight with primary antibodies diluted in TBST (Tris buffered saline with Tween-20; Genesee Scientific) with 5% milk at 4 °C, followed by incubation with secondary antibodies at room temperature for 1 h. Immunoreactivity was visualized by the ECL chemiluminescence system (Bio-rad). For MANF protein endocytosis-western blot assay, HEK293T cells were plated at a density of approximately 100,000 cells per well in a poly-L-ornithine-coated 24-well plates. Twenty-four hours after plating, cells were starved in DMEM for 24 h before 3 h of anti-sulfatide antibody (1:25) or sulfatase (20 μg/ml) treatment. Then, conditioned PBS with secreted Hs-MANF was added to the cells for indicated times of duration. Internalization of Hs-MANF were terminated by rapidly rinsing the cell three times with ice-cold PBS and lysing cells with 1× Laemmli sample buffer. The antibodies were V5 (ab3792, 1:2000; EMD Millipore Corporation, Darmstadt, Germany), HSP90 (sc-13119, 1:5000; Santa Cruz Biotechnology, TX, USA), Tubulin (T5168, 1:10,000; Cell signaling Sigma Aldrich, St. Louis, MO, USA, MA, USA), GFP (sc-9996, 1:50,001,000, Santa Cruz Biotechnology, TX, USA), His (6AT18, 1:20,003,000, Sigma Aldrich (St. Louis, MO, USA), and Anti-O4 Antibody, clone 81, MAB345, Millipore (1:100). Uncropped western blots of key experiments can be found in Supplementary Figs 8 and 9.

**Recombinant protein purification from E. coli**. The plasmid pET-28a_Hs-MANF or pET-28a_Ce-MANF (codons optimized for prokaryotic expression) was transformed in *E. coli* BL21 pLysS cells and then grown at 37 °C in LB medium containing 50 μg/ml kanamycin and to an absorbance at $\lambda = 600$ nm (A600). Protein expression was induced with 0.5 mM IPTG and cells were grown 5 h at 37 °C (Hs-MANF) or 16 h at 25 °C (Ce-MANF). Pelleted cells were resuspended in lysis buffer (potassium phosphate buffer containing 1 mg/ml lysozyme, 0.02% Triton X-100, PMSF 0.5 mM, protease inhibitor cocktail (Bimake, Houston, TX, USA), and lysed by mild sonication. After cell debris removal by centrifugation, the solubilized fraction was loaded onto a pre-equilibrated Ni$^{2+}$-nitrilotriacetate column and bound proteins were washed with washing buffer (100 mM potassium phosphate buffer, pH 7.4, containing 60 mM imidazole). The purified protein was obtained using elution buffer (potassium phosphate buffer (pH 7.4) containing 250 mM imidazole.

**Compound treatment**. The compounds used in this study were as follows: pilipin III ($C_{35}H_{58}O_{11} \geq 85\%$), Cytochalasin H ($C_{30}H_{39}NO_5 \geq 95\%$), Coelenterazine-h ($C_{26}H_{21}N_3O_2 \geq 95\%$), dynasore ($C_{18}H_{14}N_2O_4 \geq 95\%$), GW4869 (hydrochloride hydrate, ≥95%), YM-201636 ($C_{25}H_{21}N_7O_3 \geq 95\%$), obtained from Cayman Chemical Company (Ann Arbor, Michigan, USA). Thapsigargin ($C_{34}H_{50}O_{12}$, ≥95%) and sulfatase (EC: 232-772-1) were purchased from Sigma Aldrich (St. Louis, MO, USA). Sulfatides ($C_{42}H_{80}NNaO_{11}S$, bovine source, ≥98%) were purchased from Matreya LLC (State College,PA, USA). Cytochalasin H, pilipin III, Coelenterazine-h, and dynasore were added 1 h before addition of conditioned PBS and remain present in the medium. Sulfatide was added together with conditioned PBS and kept in the medium. Cytochalasin H, pilipin III, Coelenterazine-h, and dynasore were added 1 h before addition of conditioned PBS and kept in the medium. Sulfatide were added together with conditioned PBS and kept in the medium.

**Immunocytochemistry of C. elegans and mammalian cells**. For immuno-fluorescence staining of *C. elegans*, adult hermaphrodites were kept in control PBS or MANF::V5 conditioned PBS with or without sulfatide (0.5 mM) overnight. The animals were washed with M9 three times and fixed with 4% paraformaldehyde in PBS for 30 min at room temperature. The animals were then washed three times in PBS-Tween-20 (0.05% Tween-20 in PBS, pH 7.2) and incubated with 2% Tween-20 in PBS (PH 7.2) for 30 min at room temperature. The animals were rocked gently overnight at 37 °C in 5% β-mercaptoethanol, 1% Triton X-100 in 0.125 M Tris-HCl (pH 6.8). The animals were then washed and shaken vigorously in 0.4 ml of 100 mM Tris-HCl (pH 7.4), 1 mM CaCl$_2$, and 2000 U/ml collagenase type IV (Sigma, St. Lois, MO, USA). Once 20% of the animals were fragmented, the animals were washed with PBS-Tween-20 and AbA solution (1% bovine serum albumin (BSA), 0.5% Triton X-100 in PBS), and rocked overnight at room temperature in 200 μl of AbA solution containing V5 antibody (1:200) followed by additional incubation at 4 °C for 6 h. The animals were then washed three times in AbA solution and incubated for 4 h at room temperature in the dark in 200 μl of ABA solution containing Cy3-conjuated secondary antibody (1:200; EMD Millipore, Darmstadt, Germany). The animals were washed with AbA solution, mounted on a slide, and imaged by Leica (Wetzlar, Germany) confocal fluorescence microscopy.

For immunocytochemistry of mammalian cells, HEK293T cells were seeded on coverslips within 24-well plates and 24 h later, the cells were treated with Hs-MANF::V5 PBS or equivalent amount of HEK293T PBS for 24 h. The cells were fixed with 4% paraformaldehyde in PBS, washed with PBS, and permeabilized with 0.02% Triton X-100 in PBS for 10 min. Blocking was done with 5% BSA in PBS for 1 h, followed by incubation with antibodies against V5 (1:400), GRP78 (1:250) in blocking buffer overnight at 4 °C. The mouse antibodies were detected using goat anti-mouse Alexa 488 (1:1000; Jackson ImmunoResearch Laboratories Inc.) and the rabbit antibody was detected using Cy3-conjuated donkey anti-rabbit (1:500; EMD Millipore) and goat anti-mouse in blocking buffer. Cells were extensively washed with PBS after primary and secondary antibody staining. Stained cells were overlaid with Fluoroshield mounting medium with DAPI (Abcam) to label nucleus. Fluorescence microscopy was performed with a Leica confocal microscope using the following fluorescence filters: DAPI (405 nm excitation for nuclear); Cy3 (551 nm excitation) for V5 staining; and GFP (488 nm excitation). For comparison across conditions, identical light-exposure levels were used.

**Granule quantification**. The GFP-G3BP-marked U2OS cells were plated at a cell density of approximately 50,000 in 24-well plate. Twenty-four hours later, the medium was changed to serum-free DMEM to starve for 48 h. Conditioned PBS or MANF-containing medium was added to the cells 16 h before changing to medium containing 0.5 μM Thapsigargin for 40 h to induce granules. Granules were visualized by an upright laser-scanning confocal microscope (Leica SPE) for imaging and by an EVOS FL Auto 2 Cell Imaging System (Thermo Fisher Scientific Inc.) for quantification with 10 fields per time point. Magnification of ×20 was used for the analysis of the percentage of cells with granules. With ImageJ 1.48v (Wayne Rasband), the number of cells with granules and total cells per field were counted. Each microscopic field contains around 100 cells at 60% confluency. Percentages of cells with at least one granule with granules were quantified under the same criteria for the indicated sets of thapsigargin, MANF, and sulfatide treatments.

**Lipid protein overlay binding assay**. Echelon lipid overlay assay using the Sphingo strip (Catalog No.: S-6000) was performed according to the manufacturer's instruction (Echelon Biosciences Inc). For customized lipid overlay assays, sulfatide was dissolved in chloroform/methanol (1:1) at the concentration of 10 nm/μl. One microliter of the lipids was spotted onto a nitrocellulose membrane at the 1:1, 1:2, 1:4, 1:8, 1:16, 1:32, 1:64, 1:128 dilutions. The membrane was then subsequently air-dried, blocked with 3% protease-free BSA in Tris-buffered saline with 0.1% Tween (TBST) for 1 h, overlaid with His::MANF protein (0.5 μg/ml in blocking buffer) or Hs-MANF::V5 conditioned medium (1 ml conditioned medium mixed with 2 ml blocking buffer) for 1 h. After three washes with TBST, the membrane was incubated with indicated primary antibodies for 1 h or overnight at

4 °C, followed by three washes with TBST and incubating the membrane with secondary antibodies for chemiluminescence imaging.

Validated antibodies for sulfatide lipid overlay assay include anti-V5 (ab3792, dilution 1:1000; Abcam), anti-His (6AT18, 1:1000; Sigma), anti-MANF C-terminus (ab67271, dilution 1:1000; Abcam). The antibody targeting the N-terminal Saposin-like domain of MANF (ab129602, dilution 1:1000; Abcam) did not work in the sulfatide lipid overlay assay likely because of its competing with sulfatide for the same epitope. Lipids used for the lipid overlay assay include: Sulfatides (13135 and 860572, Avanti Polar Lipids; 1049, Matreya), Cerebrosides (131303 and 131304, Avanti Polar Lipids), lyso-Sulfatide (1904, Matreya), N-Dodecanoyl-NBD-sulfatide (1632, Matreya), psychosine (P9256, Sigma)

**Cell death assay.** HEK293T cells were plated in 24-well plates at a density of approximately 100,000 per each well. After 12 h of cell attachment, 10 μl concentrated PBS containing secreted Hs-MANF or Hs-MANF (K112L) in the presence or absence of sulfatide (100 μM) were added to the cells, while equal volumes of dimethyl sulfoxide were used as vehicle controls. Sixteen hours later, the cells were treated with thapsigargin (1.5 μM) for 40 h. For anti-sulfatide antibody or sulfatase pretreatment, anti-sulfatide antibody O4 (Millipore) and sulfatase (Sigma Aldrich) were added 3 h prior to MANF PBS addition. Both detached and floating cells were collected by centrifugation and washed with 1 ml PBS. The collected cells were resuspended with 300 μl PBS with addition of 0.3 μl Sytox blue (1 μM; Thermo Fisher) for an additional 5 min. The fluorescence intensity was measured for individual HEK293T cells using flow cytometry 20 min after staining, and the percentage of cell deaths was quantified using the Flowjo software.

**Sulfatide quantification in biological samples.** Cells were homogenized by sonication in MilliQ water. Total lipid fraction was obtained by Folch extraction in chloroform/methanol (2/1, v/v) mixture containing 50 ng of C12:0 sulfatide internal standard. The organic and water phases were separated by centrifugation at $14,000 \times g$ for 5 min. The lower organic phase was transferred to a glass vial and dried under stream of nitrogen (3×). Then the residue was dissolved in 500 μl of methanol prior to tandem mass-spec analysis. Mass spectra were measured on an AB/MDS ABI/MDS SCIEX API 4000 triple quadrupole tandem mass spectrometer equipped with an ESI source and coupled to an Agilent HPLC 1290 series. Samples were electrosprayed in the negative ion mode to form [M–H]− ions. Generated ions were analyzed by SRM of precursor ions and the common HSO4− ($m/z$ 97) fragment ion. Data were processed using Analyst 1.6 software.

**Biochemical and biophysical analysis of recombinant MANF.** Purified Hs-MANF was dissolved in 50 mM sodium phosphate (pH 7.4) containing 100 mM NaCl, 0.05% Tween-20, and different doses of sulfatide. Samples were analyzed using a Superdex 75 10/300 GL column when 200 μl of Hs-MANF protein samples (0.5 mg/ml) was injected. For limited proteolysis the protein samples (0.5 mg/ml) with the ligand were digested by trypsin (MANF-trypsin (w/w) ratio of 1000-1). At the chosen time point, proteolysis was quenched in 0.5% TFA. Cleaved proteins were analyzed by silver-stained SDS-PAGE. For differential scanning fluorimetry, Hs-MANF (0.5 mg/ml) was dissolved in 50 mM sodium phosphate (pH 7.4) containing 100 mM NaCl. To analyze the effect of sulfatide on protein thermostability, the proteins were incubated with the ligand in defined concentration for 10 min at room temperature prior to measurement. Using a Prometheus NT.48 (NanoTemper), the proteins were heated from 20 to 90 °C, and tryptophan-based fluorescence was recorded.

**Statistical analysis.** Data were analyzed using GraphPad Prism Software (Graphpad, San Diego, CA) and presented as means ± S.E. unless otherwise specified, with $P$-values calculated by two-tailed unpaired Student's $t$-tests or two-way ANOVA (comparisons across more than two groups) adjusted with the Bonferroni's correction. No randomization or blinding was used and no power calculations were done to detect a pre-specified effect size.

**Data availability.** The data that support the findings of this study are available from the corresponding author upon reasonable request.

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

## Acknowledgements

We thank the *Caenorhabditis* Genetics Center, National BioResource Project, and the Million Mutation Project for *C. elegans* strains. The work was supported by NIH grants R01GM117461, R00HL116654, ADA grant 1-16-IBS-197, Pew Scholar Award, Simons-Klingenstein Fellowship in Neurosciences, Packard Fellowship in Science and Engineering (to D.K.M.), a Hillblom Foundation grant (to D.K.M.), and postdoctoral fellowship (to R.V.), Grant# 15-06582S from the Czech Science Foundation (to A.H.), and the institutional program of Charles University in Prague (UNCE 204064) (to L.K.).

## Author contributions

M.B. performed mammalian cell assays, immunoblotting, *C. elegans* imaging, gathered and analyzed results. R.V. performed sulfatide binding assays, *C. elegans* genetic analysis and imaging, gathered and analyzed results. A.H. and L.K. performed MANF-sulfatide biochemical assays. C.J., T.L., Y.Z., C.W., L.F. and Y.D. contributed to MANF expression, purification and characterization. B.W. performed SNP mapping analysis. D.K.M. conceived and supervised the project, conducted *C. elegans* genetic experiments, and wrote the manuscript with significant contributions from M.B. and R.V.

## Additional information

**Competing interests:** The authors declare no competing interests.

