## [Peer Review File · Nature Communications]

Reviewers' comments:

Reviewer #1 (Remarks to the Author):

The manuscript by Bai et al uses a genetic screen to identify MANF as a regulator of ER stress, more specifically, a regulator of the HSP-4::GFP reporter in *C. elegans*. They show that human MANF expression can reduce ER stress caused by mutation in CeMANF. They provide data to support that exogenously administered MANF has cytoprotective effect in human cell culture models of ER stress and this protection is modulated by interaction with sulfatide, or 3-O-sulfogalactosylceramide. The interaction of MANF with lipids has been speculated to interact with lipids because of its N-terminal saposin-like structure but this is the first data showing an interaction between endogenous lipid structures and MANF.

Overall, the manuscript is well written with several convincing findings but inconsistencies and insufficiencies in description of the methods made some of the data difficult to compare and interpret. Moreover, many of the experiments appear to be qualitative because quantitation and statistics were not performed or, if they were, it was unclear what the “n” value or replicate number (technical/biological and experimental replicates) was used for each experiment. Specific examples and points for consideration are given below.

1) Related to Figure 1, is there data on the fecundity or health of the various crosses shown in Figure 1C? Such data from the cross may be informative as to MANF's contribution to *zc12* which has some growth and development issues (WormBase). The authors refer to XBP1 as “master regulator” of ER stress. This categorization of XBP1 is a bit misleading as it is one regulator of one pathway in the Unfolded Protein Response. Were PERK and ATF6 pathways and related mutants also examined? Consider revising to make it more specific to XBP, UPR and well-established literature base of UPR in worms. Also, consider revising “*manf-1* mutants were fully suppressed by loss of function XBP-1” to reflect that the phenotype of HSP-4::GFP upregulation is suppressed, not *manf-1* mutants effects which may extended beyond XBP1 signaling of HSP-4::GFP. Other aspects of the MANF mutation (e.g. interaction with other UPR prongs) was not examined so the term “fully suppressed” is misleading. Were all *manf-1* mutants crossed? Only *dma1* data is shown.

2) Consider providing a map of the various constructs used in the paper and where the tags are located. The specific recombinant MANF (Human, Ce, tag type, source, etc) should be clearly referenced throughout to aid in the interpretation of the results and comparisons across

experiments.

3) Figure 3a shows “purified” MANF::V5 but based on the methods, MANF was only concentrated, not purified. There is also no controls of concentrated secretome from cells with and without the sulfatide added. What is the “n” for the experiment in Fig 3c? Were the worms washed prior to WB? How does mutant MANF appear in Figure 3e? Why are different regions with different stains used for the two different tagged MANFs? Showing few different regions of the immunostained worm for both MANFs with apparently different tags would be very informative given literature on C-terminal tail of MANF and its localization. How does V5-tagged MANF compare to a control with a secreted protein (GFP that is V5 tagged for example)? Worms will pump ingested material into the oocytes and this could result in immunostaining pattern of MANF. Does the immunostaining pattern change with sulfatide since there is more MANF now internalized into the worm based on Fig3C data?

4) For Figure 4, there appears to be discrepancy with methods and figure. Figure legend says 0.5 uM for 48 hours but methods say 1.5 uM for 40 hours. No OGD/hypoxia methods are described in the paper. How does cell morphology appear after treatment? The Tg concentration and duration is relatively high and it is surprising cells aren't completely dead. What is the number “n” of wells? Number of experimental replicates? What is the source and validation of the sulfatide antibody, there is no mention in the methods.

5) The MCAO method (both filament and vessel suturing) is variable and MANF has previously been shown to improve stroke outcome in rodents in several published studies (e.g. Wang et al 2016 Int J Mol Sci, Yang et al 2014 J Neurol Sci, Airavaara et al J Comp Neurol 2009,). The WT MANF group should have minimal of same animals as others (n=5). What are the concentrations of MANF used and which tagged version? How was the concentration of MANF between WT and K112L measured? Was it same as analyzed in Sup Fig 2D? How do the concentrations used compare to published work with MANF and stroke? Based on the rest of the paper, it seems the relevant question is whether MANF pretreated with sulfatide, would it perform better? Overall, the MCAO study doesn't provide compelling novel data and could be removed and expanded on for another paper.

6) In Figure 6, how many experiments were performed? Where are the controls for Figure6C and D? What happens to pattern with mutant MANF? Or with sulfatide? The figure says “anti-MANF::V5” was a MANF antibody used or V5 antibody used? Same question for Sup Fig 6a. In general, the antibody used for staining or blots should be clearly and consistently labeled.

7) The findings in Figure 7 are an intriguing consequence of MANF treatment. Figure 7a shows difference in TG-induced G3BP staining but how does the K112 look? Or the MANF with sulfatide? How were cell boundaries defined to count granules? How many granules constituted

a positive cell? The methods state both confocal and EVOS were used, which images were used for data analysis? How many wells/independent experiments were conducted? From images shown, could the degree of cell-cell contact influence granule formation by Tg? In PBS group, the patch of cells appears to be fully contacting other cells whereas the representative image for MANF treatment shows less granules and less cell-cell contact. For Fig 7b and c, if you add sulfatide to K112L, does it have an effect? Or sulfatide alone? Lastly, given availability of MANF ELISAs or using stained protein gels with known protein standard, what is the estimated “low” and “high” concentration of MANF? This information would be useful in evaluating this finding as a possible cytoprotective mechanism in context of other published studies using recombinant MANF for cytoprotective effects.

8) The failure of the V5-tagged MANF to interact with KDEL receptors is predicted by the authors, since they use this construct to avoid ER retention. The V5-tag is masking the RTDLC-terminal tail which has been shown to be involved in ER retention and modulated by KDELRs (Henderson et al J Biol Chem 2013, Oh-Hashi Mol Cell Biochem 2012). Consider using a N-terminal tag (downstream of sigpep) instead, revise interpretation or remove. What is the source of GFP::KDEL1, no reference or methods provided. Has the GFP::KDEL1 fusion protein been shown to be functional for ER retention?

Minor

Consider using “conditioned media” instead of “conditional media” throughout manuscript.

Figure 3 b should be clearly labeled to indicate that it is manf-1 mutant (not wildtype).

Consider discussing the apparent sharp dose response of MANF binding to sulfatide (Figure 4a and C). Maybe just threshold of assay but seems very sharp given 2 fold dilution series.

In Figure 4e, the last 3 bars should all be with sulfatide but this is not clear from labels.

Consider adding brief mention of rationale/background of relevance of edaravone.

Sup Fig 5 has a MANF KO, from where? No information or reference provided.

Sup Fig 2D, is “K104L in *C. elegans*” part of 2D or 2E? What is it referring to?

In Sup Fig 6c, the signals appear to both be present but not necessarily colocalized. How frequently was this observed given there is only 1 cell shown and MANF does not appear similar to pattern shown with tagged versions of MANF in HEK293 (compare SupFig7g, Fig 6C). Consider using different tagged MANF or CST.

Reviewer #2 (Remarks to the Author):

In this study Meirong Bai and Roman Vozdek identified sulfatide as a putative lipid-based MANF receptor, in contrast to the more common mechanism of recognition by protein receptors. The authors further demonstrate that direct binding of MANF specifically by sulfatide is important for its internalization and uptake by the cells. These are important for subsequent cytoprotection from damage and cell death induced by hypoxia and by ER stress. These discoveries were based on the characterization of two homologs of MANF in two distinct systems (a newly identified *C. elegans* homolog of MANF and the well established human MANS ortholog). Yet, how MANS elicits cytoprotection once it is internalized into the cells remains illusive, and is not really addressed in this paper.

The study nicely combines genetics, cell biology and biochemical approaches to prove their case. All in all, the most significant finding of this work is the identification of an evolutionarily conserved recognition and internalization pathway, based on lipid recognition, of MANS into target cells.

In my opinion, the paper is innovative, interesting, and informative and in most cases technically sound. Although the cytoprotection mechanism is not addressed, I find that the identification of this non-canonical mode of internalization of MANS as a sufficiently exciting advancement for the field, which may be relevant for additional orphan ligands, and thus I support its publication in this prestigious journal.

Nevertheless, some points need to be further addressed and clarified before publication:

Major comments:

1) Assessment of the effect of MANS on ER homeostasis and ER stress resistance in *C. elegans*. Throughout the paper the authors claim that they have demonstrated that MANS alleviates the ER-UPR and provides cytoprotection in *C. elegans* and mammals. I accept that this claim has been well supported by the experiments performed in mammalian cells. However, the *C. elegans* experiments did not address this point (see minor comment 5 and 12).

Specifically, it was clearly demonstrated that when *C. elegans* lack MANF, at least one arm of the ER-UPR is activated (the IRE-1/XBP-1 arm). It was further shown that once you find a way to supplement these animals with MANF (by transgenic expression or by internalization of external MANF), then this ER stress response is no longer induced. However, in these experiments, the initial cause of the stress (i.e. the deficiency in MANF) is no longer there.

I would like to see an experiment assessing the vulnerability of animals to ER stress upon over-expressing or depletion of MANF.

One way to address this, at the whole animal level, is to challenge the animals with tunicamycin from eggs and follow their ability to develop into adults. Does MANF deficiency render them

more sensitive to tunicamycin? Does MANF internalization or over expression render them more resistant?

Another way to address this, is to challenge the adult animals with very high concentrations of tunicamycin, and follow apoptosis in the germline. Does MANF deficiency render them more sensitive to tunicamycin? Does MANF internalization or over expression render them more resistant?

This is an important aspect of the work, which is feasible yet currently missing.

These experiments can support the claims the MANF confers cytoprotection in *C. elegans*.

2) In many cases, experiments are presented as a single repeat or as a representative picture with no accompanying quantification and/or statistics.

For example:

a) Fig 1c, 1f, 3b– the levels of the reporter intensity should have been assessed in a quantitative way. It can be presented as % animals expressing the reporter in the intestine, or one can measure the fluorescence intensity of the reporter in many individuals of the same genetic background. The number of animals scored should be reported and a statistical analysis of the results should be presented.

b) Fig 3c, 3d– only a single western blot is presented. Is it truly representative? Can a densitometry graph accounting for the repeats of the experiments be added?

3) Figure 2F demonstrates the protection of MANF digestion by trypsin by increasing concentrations of sulfatide. I would like to see a control of another protein substrate whose digestion by trypsin is not affected by increasing amounts of sulfatide. Supp Fig 3A presents a control in the form of MANF with the K112L mutation, but I don't think the same range of sulfatide was used there. Please clarify the meaning of the ratios 1:10, 1:25, 1:50.

Minor Comments:

1) Introduction – page 3, 2 references are missing. On line 16 a reference regarding the expression pattern of MANF in mammals is missing. On line 18, a reference regarding changes in abundance and secretion of MANF upon various stresses is missing.

On the same note, on page 5 line 6 the authors claim that they isolated mutants with constitutive GFP expression even without ER or hypoxic stresses. However such stresses may have been generated by the mutations themselves. Hence it should be noted that the mutants with

constitutive GFP expression were identified even without externally induced ER or hypoxic stresses.

2) Results – page 5, 1st row – it is claimed that the screen in *C. elegans* was designed to identify genes involved in HIF-independent ER stress response, but no action was taken to ensure that the identified mutations are not related to HIF or to hypoxia. Hence, this statement should be removed.

3) Chaperon should be spelled as chaperone.

4) Page 5 line 17 – Supp Fig. 1d should also be referenced here.

5) Page 6 line 4 – The conclusion that *manf-1* mutants were fully suppressed by LOF of XBP-1 is an overstatement. Only the expression of the Phsp-4 reporter was assessed.

Page 6 line 6 – The conclusion that *C. elegans* MANF protects cell from ER stress is not well founded. The only experiments done in this context were to show that MANF depletion activates one arm of the ER-UPR in *C. elegans*. I think that whether MANF affects sensitivity to ER stress and/or affects ER homeostasis and function in *C. elegans* is important and should be addressed (as requested in major concerns #1), and are missing from the current version of the manuscript.

6) Page 6, line 9 – this section has nothing to do with cytoprotection. Please rephrase.

7) Page 6, line 13 – sup. Fig. 2a has no information about the SAP-like domain.

8) Page 7 – title on line 8 should be changed as cytoprotection by Ce-MANF was not shown.

9) Supp Fig 4 is important and should be part of the main figures.

10) Page 8, 3 lines from bottom – were instead of was.

11) Page 9, 4 lines from bottom – GRP78 is the ER resident chaperone BIP. This indicates that internalized MANF co-localizes with the ER. This should be emphasized as it might be related to its effects on the ER-UPR.

12) Page 11, 6 lines from bottom – The sentence “*C. elegans* experiments clearly indicate the roles of both Ce-MANF and Hs-MANF in alleviating ER stresses in *C. elegans*” are not justified. All that has been shown is their ability to compensate for endogenous MANF deficiency, as assessed by the UPR reporter. Their effects on other inducers of ER stress (other than the direct deficiency in *manf-1* itself) should be assessed (as requested in major concerns #1).

13) Page 12 – The first paragraph referring to Fig 7b and 7c is unclear. Is the suppression affect of the K112L mutation referring to the inability of that form of the protein to suppress granule formation? If so, I don't think that the term suppression is appropriate here.

On line 4 – it is stated that the affects of the sulfatide treatment manifested more predominantly with higher doses of MANF, but it seems to me that they manifested more when low doses of MNF were used (Fig 7B vs 7C).

Please clarify. Does this affect the rest of your interpretation on lines 5-6?

On line 7 – Once again , alleviation of ER stress in *C. elegans* has not been shown under stress conditions other than the deficiency in MANF itself.

15) The tubulin western blot in Fig 3d is overexposed.

16) Figure 3E – the two pictures are of different regions of the worm. Are the expression patterns different?

Reviewer #3 (Remarks to the Author):

In this report, Dengke Ma and colleagues show that the mesencephalic astrocyte derived neurotrophic factor (MANF) binds the sphingolipid sulfatide, and that this association promotes cytoprotection. First, the authors demonstrate that these properties are found in MANF proteins in *C.elegans* and human, suggesting that their functional properties are conserved during the evolution. Sulfatide binding promotes MANF cellular uptake and cytoprotection from apoptotic conditions such as hypoxia and ischemia.

The discovery of MANF as a sulfatide-binding protein, whose association contributes to cytoprotection, is interesting. Also, the explanation of the observed sulfatide-dependent endocytic internalization of MANF is due to the ability of sulfatides to act as MANF chaperones is quite appealing. Whereas the biological data is elegant, the biochemical experiments need additional work. I have some concerns about the results obtained with a putative sulfatide-binding mutant of MANF (K112L) that can be addressed with appropriate controls. Also, clarification of specific experimental conditions using sulfatide is warranted.

Major comments:

1. The authors indicate that “MANF also exhibited high affinity to sulfatide” (page 6). I disagree with this observation. The authors are not performing kinetic experiments or quantifying the interaction. To indicate such statement the authors should measure kinetic constants such as KD of the association.

2. Figures 2d-e: Whereas the shift at higher molecular peak suggest that the protein is forming a larger complex, an evidence that the protein is indeed in that larger MW peak is warranted. There is also a possibility that the first peak corresponds to detergent-sulfatide complex (free of protein) used for the preincubation with MANF. The authors can also use a control such as the sphingolipids 1, 2, 3, or 4 shown in Figure 2c.
3. How sulfatide was prepared for the trypsin limited proteolysis? Sulfatide is insoluble unless a detergent or a liposome is used. Thus, controls with detergent (or anything used to solubilize sulfatide) must be included to strength the indicated observations.
4. Figure 2f: A demonstration that trypsin activity is not altered by sulfatides is required to confirm the sulfatide protection of trypsin-mediated MANF cleavage.
5. Figure 3: the authors can demonstrate the enhanced effect of sulfatide in Panel 3b (right) by adding a control with another sphingolipid to show specificity in a biological setting.
6. How sulfatides were prepared and in what amounts before they were added in the liquid cultures for the experiments shown in Figure 3, panel b?
7. I have some concerns about the data obtained with MANF K112L (Figure 4). Whereas it is nice to see a poorer lipid binding and lower cytoprotective capacities by this mutant, the mutation may alter structure, not just sulfatide binding. By looking at the structure of human MANF, it is clearly observed that K112 maps at the end of helix 6, and that a mutation in this residue can potentially alter the secondary, and perhaps the tertiary structure, of the N-terminal saposin-like domain. This would result in lower binding properties as reflected in the numerous experiments performed by the authors. Therefore, a demonstration that this is not the case is needed to make the results obtained with the K112L mutant biologically relevant.
8. Fig 3S, panel b: whereas the sulfatide protected fragment includes the N-terminus of MANF, there is no evidence that this is because of sulfatide binding. Since this is just limited proteolysis, trypsin will first target disordered regions. I recommend the authors be more cautious about the statement in page 13.
9. Figure S3, panel c: I agree that the presence of sulfatide may make MANF more thermostable. However, there are missing controls. Was sulfatide dissolved in detergent? If so, a control of MANF + detergent (at a concentration for 1:50 sulfatide) must be shown. Also, it would be appealing to confirm this observation by showing any of the noninteracting sphingolipids shown in Figure 2c do not lead to a delay in unfolding of MANF at 1:50 ratio.
10. Figure 5: The observed effects are clear but these are due to a mutation in MANF that cannot

be necessarily be functional to sulfatide binding but rather structural. No data in this Figure includes the use of sulfatide. The authors can affirm the in vivo effects of this mutation if they demonstrate that the mutation alter function, not structure.

11. How sulfatide was prepared to reduce the number of cell granules shown Figure 7?

Minor comments:

1. A graphic representation of the MANF domains and critical mutations will be helpful to better understand the author's conclusions.

2. Please provide the sulfatide amount range used in Figure 4, panels a and c.

3. Page 17: Triton should be read as Triton –X100.

4. Please provide details of the antibodies in pages 19 (Immunocytochemistry) and 20 (lipid protein overlay binding assay).

5. What does TBST stand for?

6. Can the authors comment about the LPA binding ability of Hs-MANF (which is not observed in Ce-MANF)?

Responses to Reviewers:

We appreciate all three reviewers for their thoughtful and unanimously enthusiastic comments regarding the findings. Following their suggestions, we have revised the manuscript extensively to clarify many points raised by reviewers, mainly concerning the description of results and methods. In addition, we have performed substantial additional experiments (Tm sensitivity of *manf-1* mutants, endocytosis of tag-free MANF, quantification of phenotypic penetrance of mutants, specificity of sulfatide dependency of MANF cleavage by trypsin and control experiments etc) with new results added to address all of reviewers' concerns and questions. We believe the revised manuscript has been significantly clarified and improved thanks to the three reviewers. Below please see a point-to-point response:

Reviewers' comments:

Reviewer #1 (Remarks to the Author):

The manuscript by Bai et al uses a genetic screen to identify MANF as a regulator of ER stress, more specifically, a regulator of the *HSP-4::GFP* reporter in *C. elegans*. They show that human MANF expression can reduce ER stress caused by mutation in CeMANF. They provide data to support that exogenously administered MANF has cytoprotective effect in human cell culture models of ER stress and this protection is modulated by interaction with sulfatide, or 3-O-sulfogalactosylceramide. The interaction of MANF with lipids has been speculated to interact with lipids because of its N-terminal saposin-like structure but this is the first data showing an interaction between endogenous lipid structures and MANF.

Overall, the manuscript is well written with several convincing findings but inconsistencies and insufficiencies in description of the methods made some of the data difficult to compare and interpret. Moreover, many of the experiments appear to be qualitative because quantitation and statistics were not performed or, if they were, it was unclear what the "n" value or replicate number (technical/biological and experimental replicates) was used for each experiment.

Thank you for pointing out the insufficiency of our method description. We have now revised the method description extensively to clarify specific points discussed below and explicitly indicated the quantification and statistics including replicate numbers for each experiment in the revised manuscript.

Specific examples and points for consideration are given below.

1) Related to Figure 1, is there data on the fecundity or health of the various crosses shown in Figure 1C? Such data from the cross may be informative as to MANF's contribution to *zc12* which has some growth and development issues (WormBase). The authors refer to XBP1 as "master regulator" of ER stress. This categorization of XBP1 is a bit misleading as it is one regulator of one pathway in the Unfolded Protein Response. Were PERK and ATF6 pathways and related mutants also examined? Consider revising to make it more specific to XBP, UPR and well-established literature base of UPR in worms. Also, consider revising "*manf-1* mutants were fully suppressed by loss of function *XBP-1*" to reflect that the phenotype of *HSP-4::GFP* upregulation is suppressed, not *manf-1* mutants effects which may extended beyond XBP1 signaling of *HSP-4::GFP*. Other aspects of the MANF mutation (e.g. interaction with other UPR prongs) was not examined so the term "fully suppressed" is misleading. Were all *manf-1* mutants crossed? Only *dma1* data is shown.

The reviewer raises a good point since indeed *zc12* mutations can cause pleiotropic growth phenotype as annotated in Wormbase and also as we observed, likely because of contribution of XBP-1 to UPR during development (e.g. in Richardson et al., 2011 PloS Genetics). As such, we performed RNAi against *xbp-1* by the RNAi feeding method from larval stages rather than genetic crossing for all of the mutants (Fig. 1c). Nonetheless, we observed consistent activation of *hsp-4p::GFP* phenotype caused by different *manf-1* alleles and all of them can be suppressed by RNAi against *xbp-1*. We have

revised the presentation in Fig. 1 to show RNAi results, clarified this point in the text and also in the method description.

PERK and ATF6 pathways are not as well characterized as XBP-1 in *C. elegans*, and we agree that “master regulator” of ER stress response and “fully suppressed” are somewhat misleading and have removed such statements. To further support specific involvement of the XBP-1 branch in mediating *hsp-4p::GFP* activation by *manf-1* mutants, we performed new experiments and added new result in Fig. 1c showing that RNAi against *ire-1*, encoding the ER stress sensor and endonuclease for splicing of *xbp-1*, can similarly suppress *zcls4; manf-1* phenotype. Following suggestions from this reviewer to make results more quantitative, we have also added penetrance results in Fig. 1c and Fig. 1f for each genotype.

2) Consider providing a map of the various constructs used in the paper and where the tags are located. The specific recombinant MANF (Human, Ce, tag type, source, etc) should be clearly referenced throughout to aid in the interpretation of the results and comparisons across experiments.

We have made a new graphic in Supplementary Fig. 2d to illustrate the various MANF domains, tags and critical mutations used in this paper. In addition, to make the used constructs clear and easy to interpret, we now have standardized the naming of each construct so that Human and *C. elegans* His::MANF purified from *E. Coli* and human MANF::V5 used for lentiviral overexpression are labelled consistently throughout of the revised paper.

3) Figure 3a shows “purified” MANF::V5 but based on the methods, MANF was only concentrated, not purified. There is also no controls of concentrated secretome from cells with and without the sulfatide added. What is the “n” for the experiment in Fig 3c? Were the worms washed prior to WB? How does mutant MANF appear in Figure 3e? Why are different regions with different stains used for the two different tagged MANFs? Showing few different regions of the immunostained worm for both MANFs with apparently different tags would be very informative given literature on C-terminal tail of MANF and its localization. How does V5-tagged MANF compare to a control with a secreted protein (GFP that is V5 tagged for example)? Worms will pump ingested material into the oocytes and this could result in immunostaining pattern of MANF. Does the immunostaining pattern change with sulfatide since there is more MANF now internalized into the worm based on Fig3C data?

Thanks for pointing out this typing error. Fig. 3a is based on experiments using concentrated MANF::V5 not purified MANF::V5. Concentrated HEK293T PBS with and without the sulfatide was used as control without effect. We have clarified this in revised legend and also added the method description for purification of MANF::V5 from conditioned media of MANF::V5 overexpressing cell lines as the purified MANF::V5 was used in supplementary Fig. 7b. Fig. 3c has n=3 independent biological repeats and the worms were washed with M9 for three times prior to wb - we have revised the legend to clarify. We also quantified the penetrance of the rescue and presented the percentages in the revised Figure. For immunostaining, we noticed that anti-V5 but not anti-His antibody seems to stain certain unidentified endogenous antigen in *C. elegans*, thus we have removed the immunostaining with MANF::V5 and only kept His::MANF results with newly added control image with His::MANF in PBS only. Since detection of internalized MANF::V5 in WB requires exogenous sulfatide and specific endocytosis machineries (Fig. 3c and 3d), we consider it unlikely that the WB and immunostaining signals we obtained simply because “worms pump ingested materials into the oocytes” and thus we consider that MANF without sulfatide is a sufficient control against which to compare effect with sulfatide and to support our conclusion that sulfatide promotes MANF update in *C. elegans*.

Nonetheless, we performed additional new experiments to address concerns on the potential non-specific effect by V5 epitope tagging and found that tag-free MANF can be as robustly as V5-tagged MANF to be endocytosed to HEK293T cells in a sulfatide-dependent manner (Supplementary Fig. 5c), further strengthening our original conclusions.

4) For Figure 4, there appears to be discrepancy with methods and figure. Figure legend says 0.5 μ M for 48 hours but methods say

1.5 μ M for 40 hours. No OGD/hypoxia methods are described in the paper. How does cell morphology appear after treatment? The Tg concentration and duration is relatively high and it is surprising cells aren't completely dead. What is the number "n" of wells? Number of experimental replicates? What is the source and validation of the sulfatide antibody, there is no mention in the methods. **Thank you for pointing this out. We apologize for the error as 0.5 μ M and 40 hours are correct. Regarding cell morphology changes after treatment after treatment, we noted that the dying cells turn roundish and were included for analysis. Cell aren't completely dead likely because the nature of HEK293T cell line makes them relatively resistant to ER stress. We used 10 images for technical replicates in three independent biological experimental replicates. We have also added OGD/hypoxia methods as well as sulfatide antibody information in the method description of the revised manuscript.**

5) The MCAO method (both filament and vessel suturing) is variable and MANF has previously been shown to improve stroke outcome in CWT MANF group should have minimal of same animals as others (n=5). What are the concentrations of MANF used and which tagged version? How was the concentration of MANF between WT and K112L measured? Was it same as analyzed in Sup Fig 2D? How do the concentrations used compare to published work with MANF and stroke? Based on the rest of the paper, it seems the relevant question is whether MANF pretreated with sulfatide, would it perform better? Overall, the MCAO study doesn't provide compelling novel data and could be removed and expanded on for another paper.

We have indicated the exact concentrations of His::MANF protein used and how they were measured in the revised Fig. 5 and method descriptions. In Fig. 5, we focused on characterizing endogenous sulfatide binding-deficient MANF mutants and did not examine effects of MANF pretreated with sulfatide, as we reason that exogenous source of sulfatide would be metabolized quickly once injected in vivo (Takahashi and Suzuki, 2012). We agree that MANF at comparable levels has been previously shown to improve stroke outcome. However, the K112L mutant data are new and striking, supporting the importance of sulfatide binding of MANF in cytoprotection. Thus, we would like to keep the figure in the paper.

6) In Figure 6, how many experiments were performed? Where are the controls for Figure 6C and D? What happens to pattern with mutant MANF? Or with sulfatide? The figure says "anti-MANF::V5" was a MANF antibody used or V5 antibody used? Same question for Sup Fig 6a. In general, the antibody used for staining or blots should be clearly and consistently labeled.

Shown are representative images and at least three times of independent experiments were repeated. The control is conditioned PBS from parental HEK cells and the image is now included. "anti-MANF::V5" should be anti-V5 and we have revised to use "V5-stained Hs-MANF" rather than the confusing anti-MANF::V5 description. Accordingly, we have now also clearly and consistently labelled the antibody with "anti-V5" throughout the paper. The figure shown is with sulfatide addition; without sulfatide, the signal of V5 staining is weak and not as quantitative as Western blot detection of entire cell population. To make the data presentation consistent for staining and blots, we only kept the results for HEK cells.

7) The findings in Figure 7 are an intriguing consequence of MANF treatment. Figure 7a shows difference in TG-induced G3BP staining but how does the K112 look? Or the MANF with sulfatide? How were cell boundaries defined to count granules? How many granules constituted a positive cell? The methods state both confocal and EVOS were used, which images were used for data analysis? How many wells/independent experiments were conducted? From images shown, could the degree of cell-cell contact influence granule formation by Tg? In PBS group, the patch of cells appears to be fully contacting other cells whereas the representative image for MANF treatment shows less granules and less cell-cell contact. For Fig 7b and c, if you add sulfatide to K112L, does it have an effect? Or sulfatide alone? Lastly, given availability of MANF ELISAs or using stained protein gels with known protein standard, what is the estimated "low" and "high" concentration of MANF? This information would be useful in evaluating this finding as possible cytoprotective mechanism in context of other published studies using recombinant MANF for cytoprotective

effects.

We have elaborated in the revised method description the detailed procedure and scoring criteria for granule assays. Images of microscopic fields were taken under 20x magnification with each field containing around 100 cells when the cells reach 60% confluency. As each cell has different number of granules, we quantified percentages of cells with granules instead of counting how many granules each cell has. To calculate the percentage of cells with granule formation, we defined cells with at least 1 granule as type 1 marked and counted by image J, and cells without granules as type 2. So the percentage of cells with granule = $\text{type1}/\text{type1}+\text{type2}$ was determined with exactly the same sets of criteria for indicated sets of Tg, MANF and sulfatide treatment. Although cell-cell contact might influence granule formation, we seeded cell at the same density and quantified fields with comparable cell densities among different conditions.

We agree sulfatide alone would be informative but unfortunately in this assay exogenous sulfatide itself increases granule formation, likely because of the intrinsic sulfatide toxicity to the U2OS cells unless MANF::V5 is present. We have added in method description how “cell boundaries were defined to count granules” and “How many granules constituted a positive cell”. K112 results are presented in Fig. 7b and 7c and images in Fig. 7a are representative. EVOS and ImageJ were used for the data analysis and 10 microscopic fields from 3 wells/3 independent experiments were used. For the “low dose” vs “high dose”, we have performed new blot experiments to verify the low and high MANF protein levels used for experiments (Fig. 7c). We have revised accordingly to clarify these points in the paper.

8) The failure of the V5-tagged MANF to interact with KDEL receptors is predicted by the authors, since they use this construct to avoid ER retention. The V5-tag is masking the RTDL C-terminal tail which has been shown to be involved in ER retention and modulated by KDELRs (Henderson et al J Biol Chem 2013, Oh-Hashi Mol Cell Biochem 2012). Consider using a N-terminal tag (downstream of sigpep) instead, revise interpretation or remove. What is the source of GFP::KDEL1, no reference or methods provided. Has the GFP::KDEL1 fusion protein been shown to be functional for ER retention?

We agree with this reviewer that the KDELR1 result is indeed confusing and have removed it in the revised paper.

Minor

Consider using “conditioned media” instead of “conditional media” throughout manuscript.

We agree and have changed the wording throughout in the revised paper.

Figure 3 b should be clearly labeled to indicate that it is *manf-1* mutant (not wildtype).

We agree and have indicated clearly in the revised Figure.

Consider discussing the apparent sharp dose response of MANF binding to sulfatide (Figure 4a and C). Maybe just threshold of assay but seems very sharp given 2 fold dilution series.

We agree it is very likely assay sensitivity threshold since lipid concentration is linear but the antibody (1st and 2nd) binding signal is not.

In Figure 4e, the last 3 bars should all be with sulfatide but this is not clear from labels.

The last 3 bars are with exogenous sulfatide, IgG control and antibody against endogenous sulfatide as labeled in the revised Figure.

Consider adding brief mention of rationale/background of relevance of edarvarone.

We agree and have indicated clearly in the Fig. 5 and also legend of the revised paper.

Sup Fig 5 has a MANF KO, from where? No information or reference provided.

MANF KO was generated by CRISPR mediated deletion by us. We have included it in the new method description.

Sup Fig 2D, is "K104L in *C. elegans*" part of 2D or 2E? What is it referring to?

K104L is the Hs-MANF(K112L) counterpart in *C. elegans* and part of supplementary Fig. 2d in the revised version.

In Sup Fig 6c, the signals appear to both be present but not necessarily colocalized. How frequently was this observed given there is only 1 cell shown and MANF does not appear similar to pattern shown with tagged versions of MANF in HEK293 (compare SupFig7g, Fig 6C). Consider using different tagged MANF or CST.

We agree with this reviewer's assessment. Since this is not directly relevant evidence, we have removed it.

Reviewer #2 (Remarks to the Author):

In this study Meirong Bai and Roman Vozdek identified sulfatide as a putative lipid-based MANF receptor, in contrast to the more common mechanism of recognition by protein receptors. The authors further demonstrate that direct binding of MANF specifically by sulfatide is important for its internalization and uptake by the cells. These are important for subsequent cytoprotection from damage and cell death induced by hypoxia and by ER stress. These discoveries were based on the characterization of two homologs of MANF in two distinct systems (a newly identified *C. elegans* homolog of MANF and the well established human MANS ortholog). Yet, how MANS elicits cytoprotection once it is internalized into the cells remains illusive, and is not really addressed in this paper.

The study nicely combines genetics, cell biology and biochemical approaches to prove their case. All in all, the most significant finding of this work is the identification of an evolutionarily conserved recognition and internalization pathway, based on lipid recognition, of MANS into target cells.

In my opinion, the paper is innovative, interesting, and informative and in most cases technically sound. Although the cytoprotection mechanism is not addressed, I find that the identification of this non-canonical mode of internalization of MANS as a sufficiently exciting advancement for the field, which may be relevant for additional orphan ligands, and thus I support its publication in this prestigious journal. Nevertheless, some points need to be further addressed and clarified before publication:

Major comments:

1) Assessment of the affect of MANS on ER homeostasis and ER stress resistance in *C. elegans*.

Throughout the paper the authors claim that they have demonstrated that MANS alleviates the ER-UPR and provides cytoprotection in *C. elegans* and mammals. I accept that this claim has been well supported by the experiments performed in mammalian cells. However, the *C. elegans* experiments did not address this point (see minor comment 5 and 12).

Specifically, it was clearly demonstrated that when *C. elegans* lack MANF, at least one arm of the ER-UPR is activated (the IRE-1/*XBP-1* arm). It was further shown that once you find a way to supplement these animals with MANF (by transgenic expression or by internalization of external MANF), then this ER stress response is no longer induced. However, in these experiments, the initial cause of the stress (i.e. the deficiency in MANF) is no longer there.

I would like to see experiment assessing the vulnerability of animals to ER stress upon over-expressing or depletion of MANF . One way to address this, at the whole animal level, is to challenge the animals with tunicamycin from eggs and follow their ability to develop into adults. Does MANF deficiency render them more sensitive to tunicamycin? Does MANF internalization or over expression render them more resistant?

Another way to address this, is to challenge the adult animals with very high concentrations of tunicamycin, and follow apoptosis in germline. Does MANF deficiency render them more sensitive to tunicamycin? Does MANF internalization or over expression render them more resistant?

This is an important aspect of the work, which is feasible yet currently missing.

Thank you for the thoughtful suggestion. We agree it is feasible and have performed experiments to test whether MANF deficiency renders them more sensitive to tunicamycin. The new results are now presented in Supplementary Fig. 1e. In brief, the brood size of the WT and *manf-1* mutants is not different when grown under standard condition. Exposure of worms in the larval stage L4 to tunicamycin for 24 hours reduced live progeny numbers of *manf-1* mutants to a larger degree than wild type. On the other hand, exposure of worms to tunicamycin for 1 hour did not alter live progeny numbers of the WT nor *manf-1* mutants. These findings support that *manf-1* mutants are indeed sensitive to chronic ER stress.

2) In many cases, experiments are presented as a single repeat or as a representative picture with no accompanying quantification and/or statistics.

For example:

a) Fig 1c, 1f , 3b– the levels of the reporter intensity should have been assessed in a quantitative way. It can be presented as % animals expressing the reporter in the intestine, or one can measure the fluorescence intensity of the reporter in many individuals of the same genetic background. The number of animals scored should be reported and a statistical analysis of the results should be presented.

Following suggestions from this reviewer to make results more quantitative, we have added penetrance results (N≥100 animals for each genotype and treatment) in Fig. 1 and 3.

b) Fig 3c, 3d– only a single western blot is presented. Is it truly representative? Can a densitometry graph accounting for the repeats of the experiments be added?

The blot result is truly representative (from three biologically independent replicates with N≥100 animals for each condition) and very striking for Fig 3c. For visually less striking Fig 3d, we followed this reviewer's suggestion and performed densitometry quantification and presented new graph with statistics in the revised Figure.

3) Figure 2F demonstrates the protection of MANF digestion by trypsin by increasing concentrations of sulfatide. I would like to see a control of another protein substrate whose digestion by trypsin is not affected by increasing amounts of sulfatide. Supp Fig 3A presents a control in the form of MANF with the K112L mutation, but I don't think the same range of sulfatide was used there. Please clarify the meaning of the ratios 1:10, 1:25, 1:50.

To assess effect of sulfatide on trypsin cleavage per se, we performed new experiments using the protein cN-II that does not bind to sulfatide as control and found the trypsin action on cN-II was not altered by sulfatide (supplementary Fig. 3a). The ratios meant protein-ligand molar ratios, which have now been clarified in the revised paper.

Minor Comments:

1) Introduction – page 3, 2 references are missing. On line 16 a reference regarding the expression pattern of MANF in mammals is

missing. On line 18, a reference regarding changes in abundance and secretion of MANF upon various stresses is missing.

References have now been added.

On the same note, on page 5 line 6 the authors claim that they isolated mutants with constitutive GFP expression even without ER or hypoxic stresses. However such stresses may have been generated by the mutations themselves. Hence it should be noted that the mutants with constitutive GFP expression were identified even without externally induced ER or hypoxic stresses.

We agree and have re-worded as “without externally induced” as this reviewer suggested in the revision.

2) Results – page 5, 1st row – it is claimed that the screen in *C. elegans* was designed to identify genes involved in HIF-independent ER stress response, but no action was taken to ensure that the identified mutations are not related to HIF or to hypoxia. Hence, this statement should be removed.

We have removed the “HIF-independent” statement as suggested.

3) Chaperon should be spelled as chaperone.

We corrected the spelling.

4) Page 5 line 17 – Supp Fig. 1d should also be referenced here.

We have added the reference.

5) Page 6 line 4 – The conclusion that *manf-1* mutants were fully suppressed by LOF of *XBP-1* is an overstatement. Only the expression of the *Phsp-4* reporter was assessed.

Page 6 line 6 – The conclusion that *C. elegans* MANF protects cell from ER stress is not well founded. The only experiments done in this context were to show that MANF depletion activates one arm of the ER-UPR in *C. elegans*. I think that whether MANF affects sensitivity to ER stress and/or affects ER homeostasis and function in *C. elegans* is important and should be addressed (as requested in major concerns #1), and are missing from the current version of the manuscript.

We agree and have toned down the statement in the revised paper. Please also see above response to major concern #1.

6) Page 6, line 9 – this section has nothing to do with cytoprotection. Please rephrase.

We have rephrased to “To explore mechanisms of action of MANF”.

7) Page 6, line 13 – sup. Fig. 2a has no information about the SAP-like domain.

We have added information about the SAP-like domain.

8) Page 7 – title on line 8 should be changed as cytoprotection by Ce-MANF was not shown.

We have removed Ce-MANF from the title.

9) Supp Fig 4 is important and should be part of the main figures.

H9c2 cells are relevant but sulfatide alone appears to be toxic for these cells so we did not include the sulfatide treatment

result. To make the presentation consistent, we kept the H9C2 data to supplementary Fig. 4 while keeping HEK cell results in the main Fig. 4.

10) Page 8 , 3 lines from bottom – were instead of was.

We have corrected to “were”.

11) Page 9 , 4 lines from bottom – GRP78 is the ER resident chaperone BIP. This indicates that internalized MANF co-localizes with the ER. Thi should be emphasized as it might be related to its affects on the ER-UPR.

We have emphasized by “GRP78 (i.e. the ER resident chaperone BIP)”.

12) Page 11, 6 lines from bottom – The sentence “*C. elegans* experiments clearly indicate the roles of both Ce-MANF and Hs-MANF in alleviating ER stresses in *C. elegans*” are not justified. All that has been shown is their ability to compensate for endogenous MANF deficiency, as assessed by the UPR reporter. Their affects on other inducers of ER stress (other than the direct deficiency in *manf-1* itself) should be assessed (as requested in major concerns #1).

We have performed new experiments to test sensitivity of *manf-1* mutants to ER stress. Please see above response to major concern #1.

13) Page 12 – The first paragraph referring to Fig 7b and 7c is unclear. Is the suppression affect of the K112L mutation referring to the inability of that form of the protein to suppress granule formation? If so, I don't think that the term suppression is appropriate here.

We have reworded according to your suggestion to “Compared with wild type Hs-MANF, the Hs-MANF K112L mutant exhibited markedly reduced ability to attenuate stress granule formation (Fig. 7b, c)”.

On line 4 – it is stated that the affects of the sulfatide treatment manifested more predominantly with higher doses of MANF, but it seems to me that they manifested more when low doses of MNF were used (Fig 7B vs 7C).

Please clarify. Does this affect the rest of your interpretation on lines 5-6?

Thank you for pointing this out. We have corrected the statement as indeed the sulfatide treatment manifested more predominantly with low doses of MANF, likely because of saturating effects from endogenous sulfatide.

15) The tubulin western blot in Fig 3d is overexposed.

We have used a less overexposed blot and quantified the normalized levels of MANF::V5 by tubulin with statistics (Fig. 3d).

16) Figure 3E – the two pictures are of different regions of the worm. Are the expression patterns different?

To avoid confusion and make results consistently shown with WB results, we have removed the immunostaining with HIS::MANF and only kept MANF::V5 immunostaining results with controls.

Reviewer #3 (Remarks to the Author):

In this report, Dengke Ma and colleagues show that the mesencephalic astrocyte derived neurotrophic factor (MANF) binds the sphingolipid sulfatide, and that this association promotes cytoprotection. First, the authors demonstrate that these properties are found in MANF proteins in *C. elegans* and human, suggesting that their functional properties are conserved during the evolution. Sulfatide binding promotes MANF cellular uptake and cytoprotection from apoptotic conditions such as hypoxia and ischemia.

The discovery of MANF as a sulfatide-binding protein, whose association contributes to cytoprotection, is interesting. Also, the explanation of the observed sulfatide-dependent endocytic internalization of MANF is due to the ability of sulfatides to act as MANF chaperones is quite appealing. Whereas the biological data is elegant, the biochemical experiments need additional work. I have some concerns about the results obtained with a putative sulfatide-binding mutant of MANF (K112L) that can be addressed with appropriate controls. Also, clarification of specific experimental conditions using sulfatide is warranted.

Major comments:

1. The authors indicate that "MANF also exhibited high affinity to sulfatide" (page 6). I disagree with this observation. The authors are not performing kinetic experiments or quantifying the interaction. To indicate such statement the authors should measure kinetic constants such as K_D of the association.

Thank you for pointing this out. Our experiments (gel filtration, limited proteolysis, fluorescence) did not technically enable us to assess kinetics and/or K_D values. To address this question, liposome binding assay or SPR would be appropriate for future experiments. Nonetheless, we agree with this reviewer and has removed this statement concerning "high affinity" binding, without affecting our conclusions on the biological role of sulfatide binding for MANF.

2. Figures 2d-e: Whereas the shift at higher molecular peak suggest that the protein is forming a larger complex, an evidence that the protein is indeed in that larger MW peak is warranted. There is also a possibility that the first peak corresponds to detergent-sulfatide complex (free of protein) used for the preincubation with MANF. The authors can also use a control such as the sphingolipids 1, 2, 3, or 4 shown in Figure 2c.

Please consider the following points that support validity of our gel filtration experiments:

- 1) All negative controls (proteins without ligand) are performed with buffer containing detergent of the same concentration as for injected samples.**
- 2) intensity increase of 'second peak' is accompanied by a decrease of the first peak corresponding to free protein showing that peak-shifts correspond to the changes in protein complexes molecular weight.**
- 3) we performed these experiments using three different detergents (tween 20, dodecylmaltoside, dodecylphosphocholine) and the peak-shifts correspond to size of micelles for respective detergents.**
- 4) blank runs (injection of lipid only) showed no signal (i.e. no absorbance at 280nm)**
- 5) the K112L mutant exhibited lower intensity of the second peak being consistent with data from lipid overlay assay**

3. How sulfatide was prepared for the trypsin limited proteolysis? Sulfatide is insoluble unless a detergent or a liposome is used. Thus, controls with detergent (or anything used to solubilize sulfatide) must be included to strength the indicated observations. **Lipids were solubilized using detergents (Tween 20, dodecylmaltoside or dodecylphosphocholine) which were included with the same concentrations in protein buffers.**

4. Figure 2f: A demonstration that trypsin activity is not altered by sulfatides is required to confirm the sulfatide protection of trypsin-mediated MANF cleavage.

We appreciate this reviewer for raising this important point. To directly assess effect of sulfatide on trypsin cleavage per se, we performed new experiments using the protein cN-II (type II cytosolic 5'-nucleotidase) that does not bind to sulfatide as control and found the trypsin action on cN-II was not altered by sulfatide over the time course of 120 minutes (supplementary Fig. 3a). This result supports that trypsin activity is not altered by sulfatide.

5. Figure 3: the authors can demonstrate the enhanced effect of sulfatide in Panel 3b (right) by adding a control with another sphingolipid to show specificity in a biological setting.

The specificity of binding is demonstrated in Fig. 2, in which other sphingolipids show no binding.

6. How sulfatides were prepared and in what amounts before they were added in the liquid cultures for the experiments shown in Figure 3, panel b?

DMSO with 50 mM of stock solution and 500 uM final concentration. We have clarified in the revised Figure legend.

7. I have some concerns about the data obtained with MANF K112L (Figure 4). Whereas it is nice to see a poorer lipid binding and lower cytoprotective capacities by this mutant, the mutation may alter structure, not just sulfatide binding. By looking at the structure of human MANF, it is clearly observed that K112 maps at the end of helix 6, and that a mutation in this residue can potentially alter the secondary, and perhaps the tertiary structure, of the N-terminal saposin-like domain. This would result in lower binding properties as reflected in the numerous experiments performed by the authors. Therefore, a demonstration that this is not the case is needed to make the results obtained with the K112L mutant biologically relevant.

We chose this mutation because the lysine 112 residue is located on the protein surface and does not impact tertiary structure based on the solved crystal structure of MANF (Parkash et al., 2009). Lysine 112 is substituted by leucine in the nearly identical structure of CDFN, the MANF paralogue, indicating again that K112L does not alter MANF structure. In addition, structural integrity of K112L mutant is further demonstrated by gel filtration (no aggregation observed) and limited proteolysis experiments (the rate of proteolysis without lipid is not changed for K112L compared with WT MANF).

8. Fig 3S, panel b: whereas the sulfatide protected fragment includes the N-terminus of MANF, there is no evidence that this is because of sulfatide binding. Since this is just limited proteolysis, trypsin will first target disordered regions. I recommend the authors be more cautious about the statement in page 13.

We agree and have toned down the statement in the revised text since indeed protection against proteolysis could be caused by physical shielding or by stabilization of flexible segment via allosteric effects.

9. Figure S3, panel c: I agree that the presence of sulfatide may make MANF more thermostable. However, there are missing controls. Was sulfatide dissolved in detergent? If so, a control of MANF + detergent (at a concentration for 1:50 sulfatide) must be shown. Also, it would be appealing to confirm this observation by showing any of the noninteracting sphingolipids shown in Figure 2c do not lead to a delay in unfolding of MANF at 1:50 ratio.

We appreciate this point and performed control experiments (free MANF) in detergent without lipids. Moreover, results are consistent with other methods used (lipid overlay assay, gel filtration, limited proteolysis). However, experiments with non-interacting lipids indicated that they destabilized MANF in solution inducing non-specific protein precipitation.

10. Figure 5: The observed effects are clear but these are due to a mutation in MANF that cannot be necessarily be functional to

sulfatide binding but rather structural. No data in this Figure includes the use of sulfatide. The authors can affirm the in vivo effects of this mutation if they demonstrate that the mutation alter function, not structure.

We chose this mutation because the lysine 112 residue is located on the protein surface and does not impact tertiary structure based on the solved crystal structure of MANF (Parkash et al., 2009). Lysine 112 is substituted by leucine in the nearly identical structure of CDNF, the MANF paralogue, indicating again that K112L does not alter MANF structure.

11. How sulfatide was prepared to reduce the number of cell granules shown Figure 7?

Sulfatide was dissolved in DMSO and was added to the medium. We have added to revised legend and Method.

Minor comments:

1. A graphic representation of the MANF domains and critical mutations will be helpful to better understand the author's conclusions.

We have made new graphics to illustrate MANF domains, tags and critical mutations in Supplementary Fig. 2.

2. Please provide the sulfatide amount range used in Figure 4, panels a and c.

We have added the exact amount range to the revised Fig. 4.

3. Page 17: Triton should be read as Triton –X100.

We have corrected this.

4. Please provide details of the antibodies in pages 19 (Immunocytochemistry) and 20 (lipid protein overlay binding assay).

We have added the detailed antibody information in the revised Method. Here is the list of all antibodies with detailed info:

Anti-ARME1 antibody (ab67271) (Abcam, Cambridge, MA, USA) 1:1000

Anti-V5 Epitope Tag Antibody (ab3792) (EMD Millipore Corporation, Darmstadt, Germany) 1:2000

GRP 78 Antibody (A-10) (sc-376768) (Santa Cruz Biotechnology, TX, USA) 1:200

HSP 90α/β Antibody (F-8): sc-13119 (Santa Cruz Biotechnology, TX, USA) 1:1000

Anti-α-Tubulin antibody (T5168) (Sigma) 1:5000

GFP Antibody (B-2): sc-9996 (Santa Cruz biotechnology, TX, USA) 1:1000

Anti-Caspase 3 Antibody (GTX110543S), GeneTex (GeneTex, Inc. Irvine, CA, USA) 1:1000

Anti-O4 Antibody, clone 81 (MAB345), (EMD Millipore Corporation, Darmstadt, Germany) 1 : 1000

5. What does TBST stand for?

TBST stands for Tris Buffered Saline with Tween® 20, which we have clarified in revised method description.

6. Can the authors comment about the LPA binding ability of Hs-MANF (which is not observed in Ce-MANF)?

Thanks for noting this interesting observation. The *C. elegans* genome encodes an orthologue of mammalian LPAAT enzyme that catalyzes the production of PA from lysophosphatidic acid (LPA). The LPA binding ability of Hs-MANF but not Ce-MANF suggests that Hs-MANF has evolved functions in addition to sulfatide binding to perhaps modulate LPA signaling as well, which remains to be investigated in future studies.

Reviewers' comments:

Reviewer #1 (Remarks to the Author):

The authors have revised the manuscript and addressed most of the concerns previously raised. The revised manuscript is improved and the message more succinct and supported by the data with exception noted below.

The methods for OGD are still incomplete. Provide reference for method and state details of what media were used. From methods, it appears that cultures remained under no glucose conditions but were reoxygenated. Is this correct?

For Figure 5, the response does not adequately address the concerns of the n value. There is now a statement in legend of “N \geq 3 independent biological replicates”. Is that the number of animals per group? If so, the number is insufficient for the variability of the stroke model. As stated before, without combining with sulfatide and increasing the n value, the data are incomplete to provide further supporting data to the central theme of the rest of the paper. Should be removed and sent as independent follow up finding to a specialized journal.

It is not clear what statistics are being used. There is only a general statement of the statistics in methods, and for the data, only p values are given. The specific test is being used is not stated and should not need to be inferred. T-tests are often inappropriately used.

Reviewer #2 (Remarks to the Author):

In this study Meirong Bai and Roman Vozdek identified sulfatide as a putative lipid-based MANF receptor. The authors further demonstrate that direct binding of MANF specifically by sulfatide is important for its internalization and uptake by the cells. These are important for subsequent cytoprotection from damage and cell death induced by hypoxia and by ER stress.

In my opinion the paper is innovative, interesting and informative. In their revision, the authors addressed all of my previous comments. A control experiment for the trypsin activity was added and the affect of manf-1 depletion on ER stress resistance was assessed. Qualitative quantification of the penetrance of some of the phenotypes was added. Thus I feel that the concerns I have raised have been addressed.

I recommend the manuscript for publication in Nature Communication.

Reviewer #3 (Remarks to the Author):

The authors have addressed most of my (and other reviewers) questions. I have the following comments:

1. The use of detergents makes sulfatide present in micelle, monolayer arrangements. This is not physiologically relevant as membranes are arranged in lipid bilayers. Perhaps the authors should be more cautious for certain conclusions.
2. Fig. 5: The authors assume that the mutation in MANF only targets sulfatide binding. I recommend rephrasing the sentence that other effects of this mutation are also possible.
3. Please provide full name for TBST and exact composition of the buffer and list of the antibodies used and dilutions for the lipid-protein overlay assay in the Methods section.

Responses to Reviewers:

Reviewer #1 (Remarks to the Author):

The authors have revised the manuscript and addressed most of the concerns previously raised. The revised manuscript is improved and the message more succinct and supported by the data with exception noted below.

The methods for OGD are still incomplete. Provide reference for method and state details of what media were used. From methods, it appears that cultures remained under no glucose conditions but were reoxygenated. Is this correct?

Correct, we applied reoxygenation but indeed did not replace with glucose-containing media after hypoxia treatment in order to vary oxygen conditions only. We have now clarified in the method description what media were used and also added the reference using similar methods. Below is the new version for the OGD method:

“The cells were pretreated for 16 hours with His-MANF or PBS and then subjected to hypoxia and reoxygenation. For in vitro cell model of H/R, oxygen-glucose deprivation (OGD) in 293T or H9c2 cells was induced by replacing the complete high-glucose DMEM to glucose and serum-deprived medium in a hypoxia modular incubator chamber (Nuaire) saturated with 99% nitrogen. After 5 or 16 hrs of hypoxia, the cells were cultured under no glucose condition and subjected to reoxygenation under normoxia (21% O₂) for 8 hrs.”

For Figure 5, the response does not adequately address the concerns of the n value. There is now a statement in legend of “N \geq 3 independent biological replicates”. Is that the number of animals per group? If so, the number is insufficient for the variability of the stroke model. As stated before, without combining with sulfatide and increasing the n value, the data are incomplete to provide further supporting data to the central theme of the rest of the paper. Should be removed and sent as independent follow up finding to a specialized journal.

Thanks for the suggestion from the reviewer. N is the number of animals per condition. For WT group, we have 3 animals only due to a high rate of mortality of animals of this group after surgery and injection. We agree that WT MANF has been previously shown to protect in the MCAO stroke model and thus increasing numbers of animals for WT MANF might not provide new insights. Although the sulfatide-binding defective K112L mutant data are new, supporting the importance of sulfatide binding of MANF in cytoprotection in vivo, we have followed the reviewer’s suggestion to remove the figure with in vivo MCAO results, which can be independent findings for a more specialized journal.

It is not clear what statistics are being used. There is only a general statement of the statistics in methods, and for the data, only p values are given. The specific test is being used is not stated and should not need to be inferred. T-tests are often inappropriately used.

One-way ANOVA was used to analyze the statistical significance in groups. Nonetheless, we have followed the reviewer’s suggestion to remove the Fig. 5 as above.

Reviewer #2 (Remarks to the Author):

In this study Meirong Bai and Roman Vozdek identified sulfatide as a putative lipid-based MANF receptor. The authors further demonstrate that direct binding of MANF specifically by sulfatide is important for its internalization and uptake by the cells. These are important for subsequent cytoprotection from damage and cell death induced by hypoxia and by ER stress.

In my opinion the paper is innovative, interesting and informative. In their revision, the authors addressed all of my previous comments. A control experiment for the trypsin activity was added and the affect of manf-1 depletion on ER stress resistance was assessed. Qualitative quantification of the penetrance of some of the phenotypes was added. Thus I feel that the concerns I have raised have been addressed.

I recommend the manuscript for publication in Nature Communication.

Thanks for the comments from the reviewer.

Reviewer #3 (Remarks to the Author):

The authors have addressed most of my (and other reviewers) questions. I have the following comments:

1. The use of detergents makes sulfatide present in micelle, monolayer arrangements. This is not physiologically relevant as membranes are arranged in lipid bilayers. Perhaps the authors should be more cautious for certain conclusions.

We agree that the setting of the experiment does not allow us to conclude the role of membrane bound sulfatide in MANF signaling and internalization. We note the caveat of such experiments and made cautious statement in Discussion about sulfatide to chaperone MANF through direct binding, which is important for MANF internalization but could also be regulatory factor for secretion in a more physiological in vivo setting. Specific roles of membrane-bound sulfatide will need to be addressed in follow-up future studies.

2. Fig. 5: The authors assume that the mutation in MANF only targets sulfatide binding. I recommend rephrasing the sentence that other effects of this mutation are also possible.

We agree that other potential effects of the K112L mutation cannot be completely ruled out. Given this caveat and per suggestions from Reviewer 2, we have removed Fig. 5.

3. Please provide full name for TBST and exact composition of the buffer and list of the antibodies used and dilutions for the lipid-protein overlay assay in the Methods section.

The full name for TBST is Tris Buffered Saline (TBS) with Tween-20. We purchased 10x TBST from Genesee scientific (Catalogue #: 18-235B), and diluted to 1x TBST by adding 9

times of water. The composition of 1xTBST: 0.050 M Tris, 0.138 M NaCl, 0.0027 M KCl, 0.1% Tween-20, pH 7.6. We have revised the methods providing the full name of TBST and also provided more details for this assay in the methods. Below is the new version of lipid protein overlay binding assay method description:

Lipid protein overlay binding assay

Echelon lipid overlay assay using Membrane Lipid Strips (Catalog No.: P-6002) was performed according to manufacturer's instruction (Echelon Biosciences Inc). For customized lipid overlay assays, sulfatide was dissolved in chloroform/methanol (1:1) at the concentration of 10 nm/μl. 1 μl of the lipids was spotted onto a nitrocellulose membrane at the 1:1, 1:2, 1:4, 1:8, 1:16, 1:32, 1:64, 1:128 dilutions. The membrane was then subsequently air-dried, blocked with 3% protease-free Bovine Serum Albumin (BSA), in Tris buffered saline with 0.1% Tween (TBST) for 1 h, overlaid with His::MANF protein (0.5 μg/ml in blocking buffer) or Hs-MANF::V5 conditioned medium (1 ml conditioned medium mixed with 2ml blocking buffer) for 1 h. After three washes with TBST, the membrane was incubated with primary antibodies as indicated for 1 h or overnight at 4oC, followed by three washes with TBST and incubating the membrane with secondary antibodies for chemiluminescence imaging.

We have also clarified info (product catalogue number and dilutions, see also graphic summary below) about antibodies used for the lipid overlay assay. Note that antibody targeting N-terminal Saposin-like domain of MANF cannot be used in the lipid overlay assay likely because of its competing with sulfatide for the same epitope.